# Quantitative evaluation of soil anti-erodibility in the fluctuation zones of rooted soil in a large reservoir, southwest of China

**Pengcheng Wang[1,2], Xinghan Niu[1,2], Henglin Xiao[3]\*, Gaoliang Tao[1,2], Zexi Song[1,2]**

**1** Hubei Key Laboratory of Environmental Geotechnology and Ecological Remediation for Lake and River, Hubei University of Technology, Wuhan, China, **2** Key Laboratory of Intelligent Health Perception and Ecological Restoration of Rivers and Lakes, Ministry of Education, Hubei University of Technology, Wuhan, China, **3** State Key Laboratory of Precision Blasting, Jianghan University, Wuhan, China

\* xiaohenglin@hbut.edu.cn

## Abstract

The quantitative analysis of key factors influencing the erosion resistance characteristics of colluvial zone soil is a prerequisite for accurately assessing the erosion resistance ability of the soil. Therefore, this study focuses on the reservoir erosion zone of the Guanyinyan Reservoir area in the Jinsha River Basin, which is a large hydropower station. The physicochemical characteristics of the colluvial zone soil (bulk density, moisture content, total porosity, soil texture, pH, organic matter content, and aggregate stability) as well as erosion resistance capabilities (soil erodibility factor K and shear strength) with variations in water level elevation (low, middle, and high elevations) were analyzed. This study quantitatively evaluated the relative importance of soil physicochemical characteristics to soil erosion resistance, identified key influencing factors, and subsequently constructed a comprehensive evaluation model for soil erosion resistance. The research results indicate that: 1) Redundancy analysis (RDA) and correlation analysis reveal that the soil erodibility factor K is significantly negatively correlated ($P < 0.01$) with total porosity, sand content, organic matter, mean weight diameter (MWD), geometric mean diameter (GMD), water-stable aggregates larger than 0.25 mm ($WSA_{0.25}$), and dry-sieved aggregates larger than 0.25 mm ($DSA_{0.25}$). It is also significantly positively correlated ($P < 0.01$) with percentage of aggregate destruction for aggregates larger than 0.25 mm (PAD), the silt content, and the clay content. However, it was not significantly correlated with the bulk density, moisture content, or pH. The soil shear strength is significantly negatively correlated ($P < 0.05$) with the moisture content, clay content, and soil erodibility factor K. The shear soil strength is significantly positively correlated ($P < 0.05$) with the MWD and $DSA_{0.25}$. 2) Fourteen erosion resistance indicators of the colluvial zone soil in the Guanyinyan Reservoir area were selected, and a comprehensive evaluation model for soil erosion resistance was established on the basis of Principal Component Analysis (PCA). 3) The Comprehensive Soil Erosion Index (CSEI) in the Jinping

**Data availability statement:** All relevant data are within the paper and its Supporting information files.

**Funding:** This research was supported by the National Natural Science Foundation of China (No. 42307256), the Joint Funds of the Nature Science Foundation of Hubei Province (No. 2022CFD172), the Joint Funds of the National Nature Science Foundation of China (U22A20232), the Innovation Research Group Project of the Hubei Provincial Department of Science and Technology (2025AFA020), the Open Project Funding of Hubei Key Laboratory of Environmental Geotechnology and Ecological Remediation for Lake & River (HJKFYB202405), the Innovation Research Team Project of the Hubei Provincial Department of Science (No. T2024006) and the Innovation Demonstration Base of Ecological Environment Geotechnical, Ecological Restoration of Rivers and Lakes (2020EJB004).

**Competing interests:** The authors declare that they have no competing interests.

Gaunyinyan Reservoir erosion zone varies between 0.082 and 0.942 with changes in water level elevation. For different elevations, the comprehensive soil erosion indices are as follows: high (root zone soil) <middle (root zone soil) <high (un-rooted zone soil) <middle (un-rooted zone soil) <low (root zone soil) <low (un-rooted zone soil). At the same water level elevation, with decreasing flooding time, the CSEI of the un-rooted zone soil in the erosion zone increased by 41.03%, 96.91%, and 353.13% compared with that of the root zone soil. In the reservoir erosion zone of the Guanyinyan Reservoir area, the overall Comprehensive Soil Erosion Index (CSEI) decreases with increasing water level elevation. At the same elevation, the CSEI of the un-rooted zone soil is significantly greater than that of the root zone soil, and this difference further increases with decreasing flooding time.

## Introduction

Soil erosion refers to the process by which soil or other surface materials are damaged, worn away, transported, and deposited due to natural or human factors. This type of erosion mainly includes water erosion, wind erosion, gravity erosion, and freeze–thaw erosion. This process is typically caused by factors such as precipitation, wind, terrain slope, soil texture, and human activities. Soil erosion can lead to problems such as decreased soil fertility, reduced arable land, ecological damage, water pollution, and sediment accumulation, which in turn affect agricultural production and the ecological balance [1].

Soil erodibility is the vulnerability of soil particles to detachment and loss by erosive agents. An assessment of soil erodibility is needed for predicting soil loss, which describes the resistance of soil to hydraulic flushing within riparian zones [2]. The K factor in the Universal Soil Loss Equation (USLE) is a classical empirical soil erosion model in quantification the soil erodibility [3], which has been widely used currently with a careful modification [4]. Owing to the high economic costs of measuring soil erosion and the difficulty of conducting large-scale and long-term monitoring, researchers have developed various models to estimate soil erosion at regional and global scales. These models are mainly divided into empirical/statistical models (such as the Revised Universal Soil Loss Equation (RUSLE)) and physical models (such as the Water Erosion Prediction Project (WEPP) model and the Pan-European Soil Erosion Assessment (PESERA) model) [5,6]. The effectiveness of physical models has been validated in many soil erosion studies [7], but driving these physical models requires the support of a large amount of data, which can be challenging to obtain in certain specific fields and long-term research processes [8]. In contrast, empirical/statistical models, especially the RUSLE model, have been widely applied for long-term assessments of local and global soil erosion [9]. In the past decade, the performance of the (R) USLE model has been fully validated [10]. These methods are also applicable for evaluating the erosion of soil in riparian zones.

Reservoir riparian zones, formed by rhythmic changes in water level after the construction and operation of hydroelectric dams, play an irreplaceable role in

biogeochemical processes within terrestrial and aquatic ecosystems and maintain the dynamic balance and ecological safety of the ecosystem [11]. Circulating hydraulic scouring in riparian zones, with frequent intensive disturbance, leading to obvious vegetation degradation, results in the significant heterogeneity of soil properties vertically, such as soil water content, soil texture, organic matter, nutrient and metal content, aggregate stability and soil shear strength, thus make the reservoir riparian zones extremely susceptible to erosion [12]. Soil erodibility in riparian zones increases significantly because of frequent changes in water pressure, leading to a rapid reduction in channel bank stability, which is the most severe environmental challenge in the ecological restoration of riparian ecosystems [13]. Therefore, it is essential and imperative to decrease soil erodibility and increase slope stability in reservoir riparian zones. In addition, soil erodibility can also be reflected and strongly influenced by some soil physicochemical properties, such as soil organic matter content, moisture content, particle size, cohesion and aggregate stability. Therefore, quantifying soil erodibility via local observations of soil properties and clarifying its influencing variables are essential for erosion risk assessment, erosion control, and ecological restoration.

It has been shown that plant roots play a much more important role in reducing soil detachment. The root is a key hub for sensing changes in the soil environment and maintaining soil stability, forming root–soil complexes through interweaving and entanglement, bonding with soil, and performing biochemical and other processes [14]. Other researchers have studied different types of plant root-soil composites and reported that the cohesive force of root-soil composites is generally greater than that of plain soil. The roots mainly achieve soil stabilization and slope protection by improving the cohesive force and have a relatively small effect on the internal friction angle. In addition, plant roots can increase soil permeability, improve nutrient activity, and enhance soil vegetation performance. Plants improve soil ecological functions mainly by affecting soil organic matter, total phosphorus, and polyphenol oxidase. In addition, the root exudates released by plant roots into the soil are not only important sources of soil organic matter, but also promote plant growth by activating nutrients. Thus, soil erodibility and physicochemical properties are significantly affected by roots. In addition, the effects of plant roots on soil properties also differ with respect to various root characteristics and environmental conditions. Under alternating dry and wet conditions, plant root systems adapt to environmental changes by altering their morphological structure. For example, plants increase the total root length, root surface area, and number of root tips to expand the area for water absorption. Additionally, the root branching pattern changes, shifting from a forked branching pattern to a fishtail branching pattern. This structure reduces internal competition within the roots and improves the efficiency of water and nutrient uptake. The presence of roots enhances soil permeability, increases soil infiltration rates, prolongs the runoff concentration time, and reduces surface runoff, thereby weakening the erosive force of runoff. This helps reduce soil erosion and strengthens the resistance of the soil to erosion [15].

Variations in the soil erodibility and physicochemical properties associated with dam construction have become a key focus of river studies because of the rapid expansion of hydropower development [16]. As an important form of water resource utilization, cascade reservoirs have been constructed on the Jinsha River from 2011 to 2022, including the Guanyinyan Reservoir, the vegetation restoration of which is affected not only by periodic off-season water level fluctuations but also by the climate conditions of high temperature and drought in the dry and hot valley. Currently, no large-scale research has been conducted on the Guanyinyan Reservoir area. Thus, it is essential to study the soil erodibility and physicochemical properties of the fluctuation zones of the Guanyinyan Reservoir, which play a vital role in the eco-restoration of the reservoir riparian zones of the Jinsha River and even the Yangtze River.

The response of soil detachment to different combinations of soil and root system properties has not been fully quantified. More research is needed in this area. All the aforementioned changes in near-surface soil characteristics triggered by water level fluctuations in the WLFZ probably influence soil erodibility; however, the quantitative effects and their dominant factors remain unknown. It is speculated that root traits may mediate the impact of plant diversity on riparian soil stability. To test this hypothesis, this study aimed to (1) quantify the influences of water level changes and roots on soil erodibility; (2) reveal the quantitative changes in soil erodibility with inundation duration by estimating a comprehensive erosion

index;(3) determine the predominant factors contributing to the changes in soil erodibility indicators in the reservoir riparian zones of Guanyinyan Reservoir.

## Materials and methods

### Overview of the study area

This study was carried out in the fluctuation zone of the Guanyinyan Reservoir (N26°34′09″-26, 36, 01; E 100°24′30″-100, 26, 13), which located in the Jinsha River, the upstream of the Yangtze River, northwest of China. The Guanyinyan Reservoir is the final cascade hydropower station in the 'One Reservoir, Eight Cascades' hydropower development plan in the middle reaches of the Jinsha River hydropower base. It is located at the border of Huaping County in Yunnan Province and Panzhihua City in Sichuan Province It is upstream of the Ludi La Hydropower Station and is 27 kilometers downstream from Panzhihua City. The normal reservoir water level of the power station is 1134 meters, with a storage capacity of approximately 2.072 billion cubic meters. It has an installed capacity of 300 (5 × 60) megawatts. When operated independently, it ensures an output of 478,000 kilowatts, with an annual power generation of 12.24 billion kilowatt-hours and an annual utilization time of 4080 hours. After the upstream Longpan Reservoir is put into operation, the guaranteed output increases to 1,392,800 kilowatts, and the annual power generation reaches 13.622 billion kilowatt-hours. The majority of the soil in the study area is red loam and yellow soil. The vegetation types in the reservoir erosion zone of the reservoir area are mainly herbaceous plants. The area in front of the reservoir is most significantly affected by water level fluctuations. Huaping County is the first county in the Guanyinyan Reservoir area, and the sample plots along the Jinsha River mainstream in Huaping County are typical and representative.

### Plot setup and soil collection and treatment

In July 2022, three different elevations were delineated within the reservoir erosion zone of the Guanyinyan Reservoir in the Jinsha River Basin. The boundaries were defined using the lowest water level of the Guanyinyan Reservoir at 1122.180 m and the highest water level at 1134.380 m.(http://www.cjh.com.cn/), The plots were set up at equal intervals for low (1122.2～1126. 9 m), middle (1126.9～1130. 1 m), and high (1130.1～1134. 4 m) elevations, as shown in Fig 1. At each elevation, three sampling points were evenly selected for sampling. For each plot, 9 sampling points were set up in an S-shaped belt. The collected samples were subsequently taken back to the laboratory for analysis of their physicochemical properties.

The plant species in the plots were mainly *Cynodon, Alternanthera sessilis*, and *Siberian cocklebur*. The dominant species is *Cynodon*. *Cynodon* was distributed in plots at low, middle, and high elevations, *Alternanthera sessilis* was distributed mainly in plots at low and middle elevations, and *Siberian cocklebur* was distributed only in plots at high elevations. The distributions of vegetation in plots at different elevations are shown in Table 1. Considering the species advantage of Cynodon, root samples at different elevations were taken back to the laboratory, and a scanner (Model: Epson 12000XL) was used to measure the plant root-related data, as shown in Table 2.

The soil physicochemical properties were determined via the following methods: the soil bulk density, moisture content, and total porosity were measured via the ring knife method and oven-drying method. The soil shear strength was determined through in-situ shear experiments via a portable cross-board shear apparatus (Model: SZB-1.0). The mechanical composition of the soil was analyzed via a laser particle size analyzer (Model: SCF-108A). The soil pH was measured via a pH meter (Model: PHSJ-3F). The organic matter content was determined via the potassium dichromate method with heating. Each soil sample was sieved through a 0. 25 mm mesh, and then a mixture of 5 ml 0.8 mol/L potassium dichromate and 5 ml concentrated sulfuric acid was added for sieving. Each sample was measured three times, and the results were calculated as the average of all the values. Soil erodibility factor: Boersma (1984) and others proposed a formula for estimating the K factor using only the geometric mean diameter (GMD) of the soil [17]. The soil erodibility coefficient was calculated using formulas [18]:

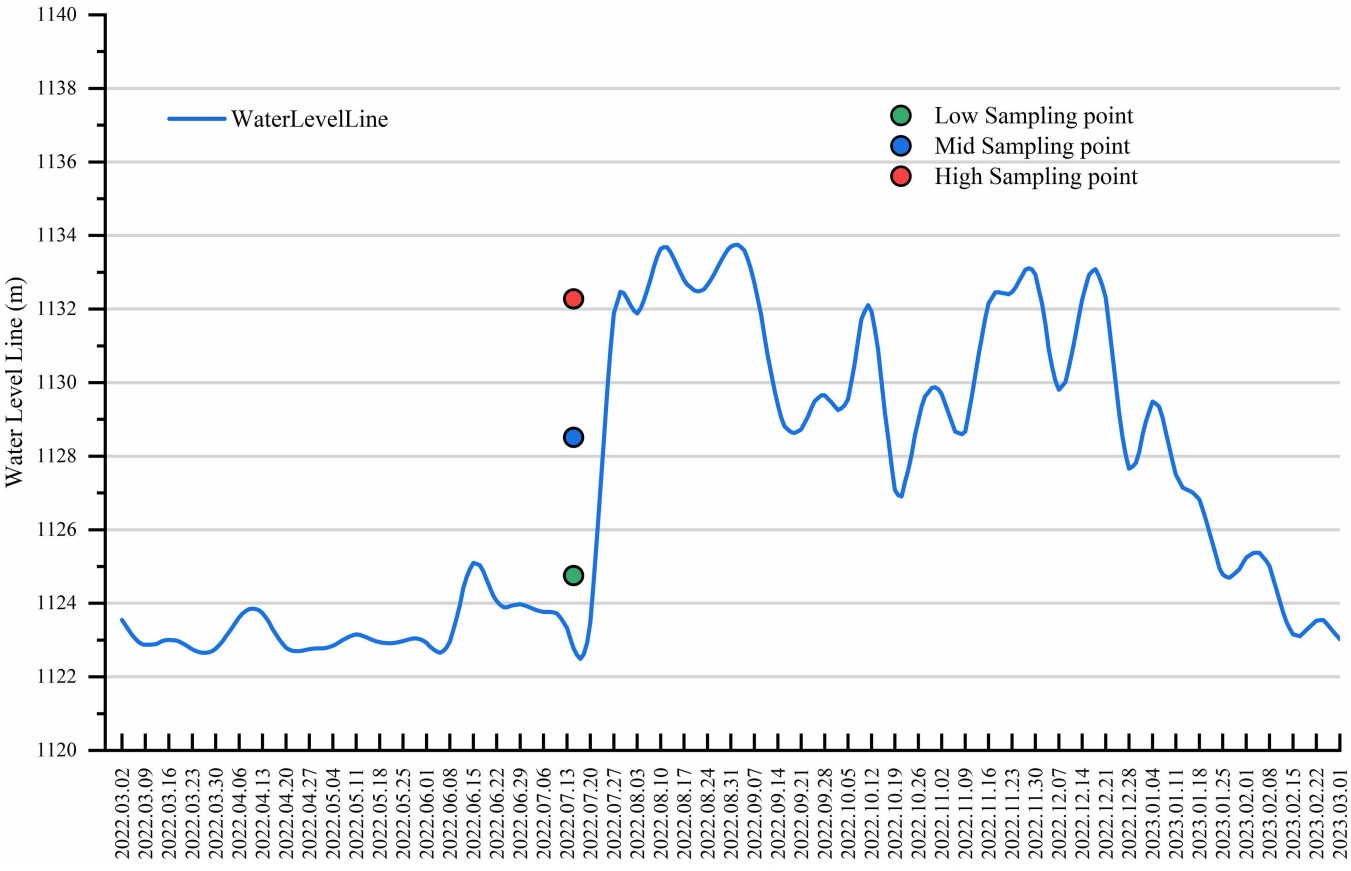

**Fig 1. Changes in water level in the Guanyinyan Reservoir area in 2022.**

**Table 1. Basic information of sample sites at different elevations in the subsidence zone of Reservoir.**

| Elevation | Slope (°) | Aspect | Coverage | Species |
|-----------|-----------|--------|----------|---------|
| Low | 3.4 | N | 32% | *Cynodon, Alternanthera sessilis* |
| Mid | 5.8 | N | 55% | *Cynodon, Alternanthera sessilis* |
| High | 8.3 | N | 78% | *siberian cocklebur* |

**Table 2. Root morphological indicators at sampling points with different elevations in the reservoir subsidence area.**

| Elevation | length (cm) | surface area (cm²) | Mean root diamete (mm) | Total Root Volume (cm³) |
|-----------|-------------|--------------------|------------------------|-------------------------|
| Low | 17.252±1.555c | 2.02814±0.592c | 0.3607±0.079b | 0.0212±0.004b |
| Mid | 20.979±2.590b | 3.3460±0.615b | 0.4111±0.052ab | 0.0455±0.008a |
| High | 27.213±2.759a | 3.9884±0.240a | 0.4775±0.074a | 0.0470±0.002a |

Different lowercase letters indicate that there are significant differences at the 0.05 level in different elevations.

$$K_i = 7.594 \left\{ 0.0017 + 0.0494 \times exp \left[ -0.5 \times \left( \frac{\log GMD + 1.675}{0.6989} \right)^2 \right] \right\}$$

When calculating $\log GMD$, the base number should be 10.

The estimated soil erodibility factor K was compared with field observations, and linear regression was used to adjust these methods to eliminate bias in the original estimates. Therefore, this study adopted the improved soil erodibility factor K calculation method proposed by Zhang et al. [19], where the difference between the calculated values and the actual values is negligible:

$$K = -0.00911 + 0.55066 K_i$$

where $Ki$ is the uncorrected erodibility factor and $GMD$ is the geometric mean diameter. $K$ denotes the actual erodibility factor. $(t \cdot h \cdot MJ^{-1} \cdot mm^{-1})$.

Soil aggregates were assessed using dry sieving and wet sieving methods [20]. The indicators for soil aggregates and stability characteristics included water-stable aggregates larger than 0.25 mm ($WSA_{0.25}$), mean weight diameter (MWD), geometric mean diameter (GMD), and percentage of aggregate destruction for aggregates larger than 0.25 mm (PAD). The calculation formulas are as follows:

$$MWD = \frac{\sum_{i=1}^{n} (R_i \times W_i)}{\sum_{i=1}^{n} W_i}$$

$$GMD = exp \left[ \frac{\sum_{i=1}^{n} (W_i \times \ln R_i)}{\sum_{i=1}^{n} W_i} \right]$$

In the equations:

MWD represents the mean weight diameter (mm).

GMD represents the geometric mean diameter (mm).

$R_i$ represents the average diameter of aggregates for the $i$ th size fraction (mm).

$W_i$ represents the weight fraction of aggregates for the $i$ th size fraction as a percentage of the soil dry weight.i corresponds to different size fractions.

$$PAD = \frac{DSA_{0.25} - WSA_{0.25}}{DSA_{0.25}} \times 100\%$$

In the equations:

PAD represents percentage of aggregate destruction for aggregates larger than 0.25 mm (%).

$DSA_{0.25}$ represents dry-sieved aggregates larger than 0.25 mm ($wt\%$).

$WSA_{0.25}$ represents water-stable aggregates larger than 0.25 mm ($wt\%$).

### The selection of soil erosion resistance indicators

Based on previous research, we selected three major categories comprising 14 soil erosion resistance indicators (Table 3).

### Data analysis

The study employed a one-way analysis of variance (ANOVA) method to detect differences in surface characteristics of soils and soil erosion resistance indicators at different water level elevations. Pearson correlation analysis and

**Table 3. Commonly used soil erosion resistance evaluation indicators.**

| Indicators of physical properties | Chemical indicators | Aggregate class |
|---|---|---|
| Soil bulk density(X1) | Organic Matter (X7) | Average weight diameter (X8) |
| Total porosity(X2) | | Geometric mean diameter (X9) |
| Moisture content(X3) | | $WSA_{0.25}$ (X10) |
| 0.05-2 mm sand content(X4) | | $DSA_{0.25}$(X11) |
| 0.002-0.05 mm silt content(X5) | | PAD (X12) |
| < 0.002 mm clay content (X6) | | K(X13) |
| | | Shear strength(X14) |

Concentrate: $WSA_{0.25}$: water-stable aggregates larger than 0.25 mm; $DSA_{0.25}$: dry-sieved aggregates larger than 0.25 mm; PAD: percentage of aggregate destruction for aggregates larger than 0.25 mm; K: soil erodibility X (1–14) is its serial number.

Redundancy analysis (RDA) methods were used to analyze the relationships between soil erosion resistance indicators and their influencing factors. To comprehensively assess soil erosion resistance, a Comprehensive Soil Erosion Index (CSEI) was established using a weighted sum method [21].

$$CSEI = \sum_{i=1}^{n} W_i U_i$$

(1)

In the above equation, CSEI represents the Comprehensive Soil Erosion Index, $W_i$ denotes the weight assigned to a specific soil erosion resistance indicator, $U_i$ represents the score of the soil erosion resistance indicator.

Analyze the variation characteristics of each indicator at different water level elevations. Evaluating soil erosion resistance using a single indicator has a certain degree of partiality and randomness. Analyzing each indicator individually is not only cumbersome but also results in some information overlap among the indicators. There is a certain degree of correlation between them, and it cannot accurately reflect the essence of soil erosion resistance. Therefore, $W_i$ Using the principal component analysis method to estimate the weights of soil erosion resistance indicators. This analysis was conducted through KMO and Bartlett's test (Table 4), and the KMO sampling suitability scale was 0.647, which is greater than the standard value of 0.6. The significance of Bartlett's test is 0, indicating a significant correlation between the secondary indicators in the study dataset. As shown in Table 5, the selected 14 erosion resistance indicators were optimized into three principal components Y1, Y2, and Y3. From the gravel plot (Fig 2), it can be seen that the eigenvalues of the principal components are all greater than 1, indicating that these three principal components can represent most of the information about soil erosion resistance in the region. Therefore, Y1, Y2, and Y3 are selected as the main components, among which Y1 includes six factors [TP(X2), MWD(X8), GMD(X9), $WSA_{0.25}$(X10), PAD(X12), K(X13)], Y2 includes two factors [Water content(X2), Shear strength(X14)], and Y3 includes six factors [BD(X2), Sand(X4), Slit(X5), Clay(X6), OM(X7), $DSA_{0.25}$(X11)]. The cumulative contribution rate of variance for the three principal components is 85.143%. The contribution rates of the three principal components are in the order of $Y_1$(59.782%)

**Table 4. KMO test and Bartlett's test of sphericity.**

| KMO Sampling Adequacy Measure | | 0.647 |
|---|---|---|
| Bartlett's test of sphericity | Approximate chi square | 332.68 |
| | freedom | 66 |
| | significance | 0 |

**Table 5. Total variance analysis of soil erosion resistance indexes in different elevation extinction zones in Guanyinyan Reservoir area.**

| Principal component | Initial eigenvalue | | |
|---|---|---|---|
| | Characteristic roots | Contribution rate (%) | Accumulated contribution rate (%) |
| 1 | 8.362 | 59.728 | 59.728 |
| 2 | 2.546 | 18.187 | 77.915 |
| 3 | 1.012 | 7.228 | 85.143 |
| | Extraction of sum of square and load | | |
| | Characteristic roots | Contribution rate (%) | Accumulated contribution rate (%) |
| 1 | 8.362 | 59.728 | 59.728 |
| 2 | 2.546 | 18.187 | 77.915 |
| 3 | 1.012 | 7.228 | 85.143 |

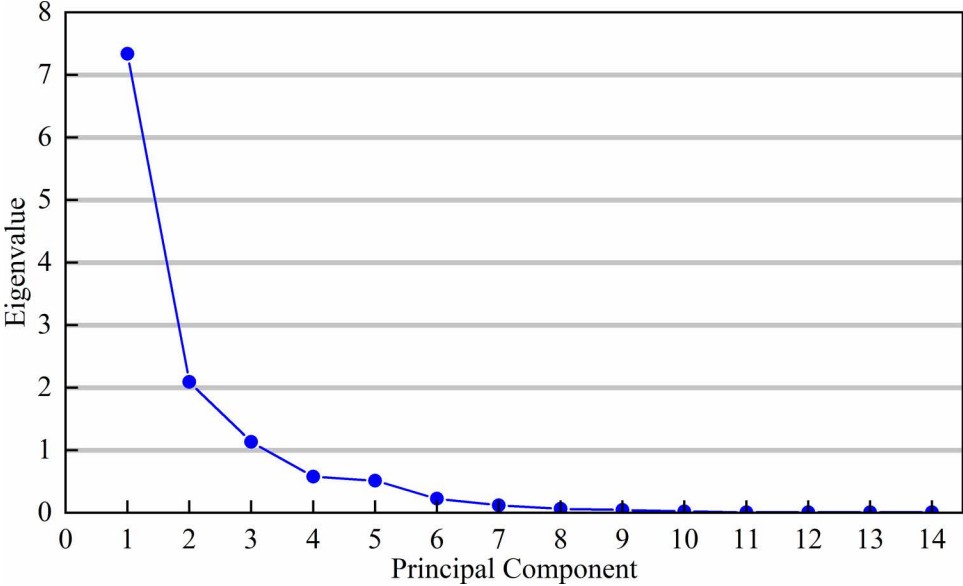

**Fig 2. Gravel chart of the soil at different water level elevations.**

> $Y_2$(18.187%) > $Y_3$(7.288%). This suggests that $Y_1$ has the greatest impact on soil erosion resistance, while $Y_3$ has the smallest impact. Using the variance contribution rates $\lambda_i$ ($i = 1,2,3$) corresponding to each principal component as weights, the equation for the comprehensive principal component index of soil erosion resistance is obtained: $Y = 0.702Y_1 + 0.214Y_2 + 0.084Y_3$.

The weight values are the ratio of the percentage of each soil erosion resistance indicator to the total percentage of variability explained by all 14 soil erosion resistance indicators Table 6. Here, n represents the number of soil erosion resistance indicators (which is 14 in this study).

$U_i$ The calculation was performed using the linear regression method with either the "S" or "reverse S" membership functions. Because BD, Water content, Sand, Silt, Clay, PAD, and K factors show a positive correlation with soil erosion resistance, they were calculated using the "S" function. In contrast, other indicators (TP, OM, MWD, $WSA_{0.25}$,

**Table 6. Score factor and weights of different soil erodibility indicators.**

| Items | BD | TP | W | Sa | Si | CI | OM |
|---|---|---|---|---|---|---|---|
| Score | 0.01 | 0.22 | 0.16 | 0.09 | 0.08 | 0.17 | 0.17 |
| Weights | 0.01 | 0.09 | 0.06 | 0.03 | 0.03 | 0.07 | 0.07 |
| Items | MWD | GMD | WSA$_{0.25}$ | DSA$_{0.25}$ | PAD | K | S |
| Score | 0.27 | 0.25 | 0.24 | 0.23 | 0.23 | 0.26 | 0.19 |
| Weights | 0.10 | 0.10 | 0.09 | 0.09 | 0.09 | 0.10 | 0.07 |

DSA$_{0.25}$, Shear strength, GMD) exhibit a negative correlation with soil erosion resistance; hence, they were calculated using the "reverse S" function. For details, refer to Table 7. $W_iU_i$ reflects the contribution of the indicator to soil erosion resistance [22].

The "S" function can be described by the membership function (7):

$$u(x) = \begin{cases} 1 & x \geq b \\ \frac{x-a}{b-a} & a < x < b \\ 0 & x \leq a \end{cases}$$

In the equation, u(x) is the membership function, x is the value of the soil erosion resistance indicator, and a and b are the lower and upper limits of the soil erosion resistance indicator, respectively (Table 7).

The reverse "S" function can be described by the membership function (8):

$$u(x) = \begin{cases} 1 & x \leq b \\ \frac{x-a}{b-a} & a > x > b \\ 0 & x \geq a \end{cases}$$

In the equation, u(x) is the membership function, x is the value of the soil erosion resistance indicator, and a and b are the upper and lower limits of the soil erosion resistance indicator, respectively (Table 7).

All data were statistically analyzed using SPSS (26.0), Excel (2013), and Canoco 5 software.

## Ethical statement

This was a purely observational study conducted in open settings. Throughout the research process, the investigators did not intervene in or influence the normal activities and environment of the subjects in any way. All observational data were collected through on-site sampling, with strict measures taken to ensure that no personally identifiable information (such as facial features, names, ID numbers, license plate numbers, etc.) was recorded. In accordance with relevant academic

**Table 7. The critical values for different indicators in the 'S' and reverse 'S' membership functions.**

| Items | Reverse S | | | | | | |
|---|---|---|---|---|---|---|---|
| | TP | OM | MWD | WSA$_{0.25}$ | DSA$_{0.25}$ | S | GMD |
| a | 55.94 | 31.53 | 2.70 | 0.80 | 0.94 | 119.33 | 1.27 |
| b | 36.29 | 7.28 | 0.62 | 0.40 | 0.86 | 12.33 | 0.27 |
| | S | | | | | | |
| | BD | W | Sa | Si | CI | PAD | K |
| a | 1.24 | 10.95 | 53.01 | 37.69 | 2.05 | 0.15 | 0.01 |
| b | 1.77 | 19.50 | 59.26 | 44.15 | 3.51 | 0.53 | 0.06 |

norms in China and the research ethical guidelines of Hubei University of Technology, such non-interventional observational research in public spaces that does not involve sensitive personal information does not require specific ethical review approval.

## Results

### Characteristics of soil physicochemical properties in different altitudinal decline zones at various water levels

Fig 3 shows that the inundation time and plant roots significantly affected basic soil characteristics in the fluctuation zone of the Guanyinyan Reservoir. The measured soil bulk density ranged from 1. 23~1.77 g·cm⁻³ (Fig 3-A). The soil bulk density has no obvious trend with the change in water level. At each elevation, at low and high elevations, the bulk density of the root zone soil was significantly lower than that of the un-rooted zone soil (P<0.05), with differences of 24.39% and 24.86%, respectively.

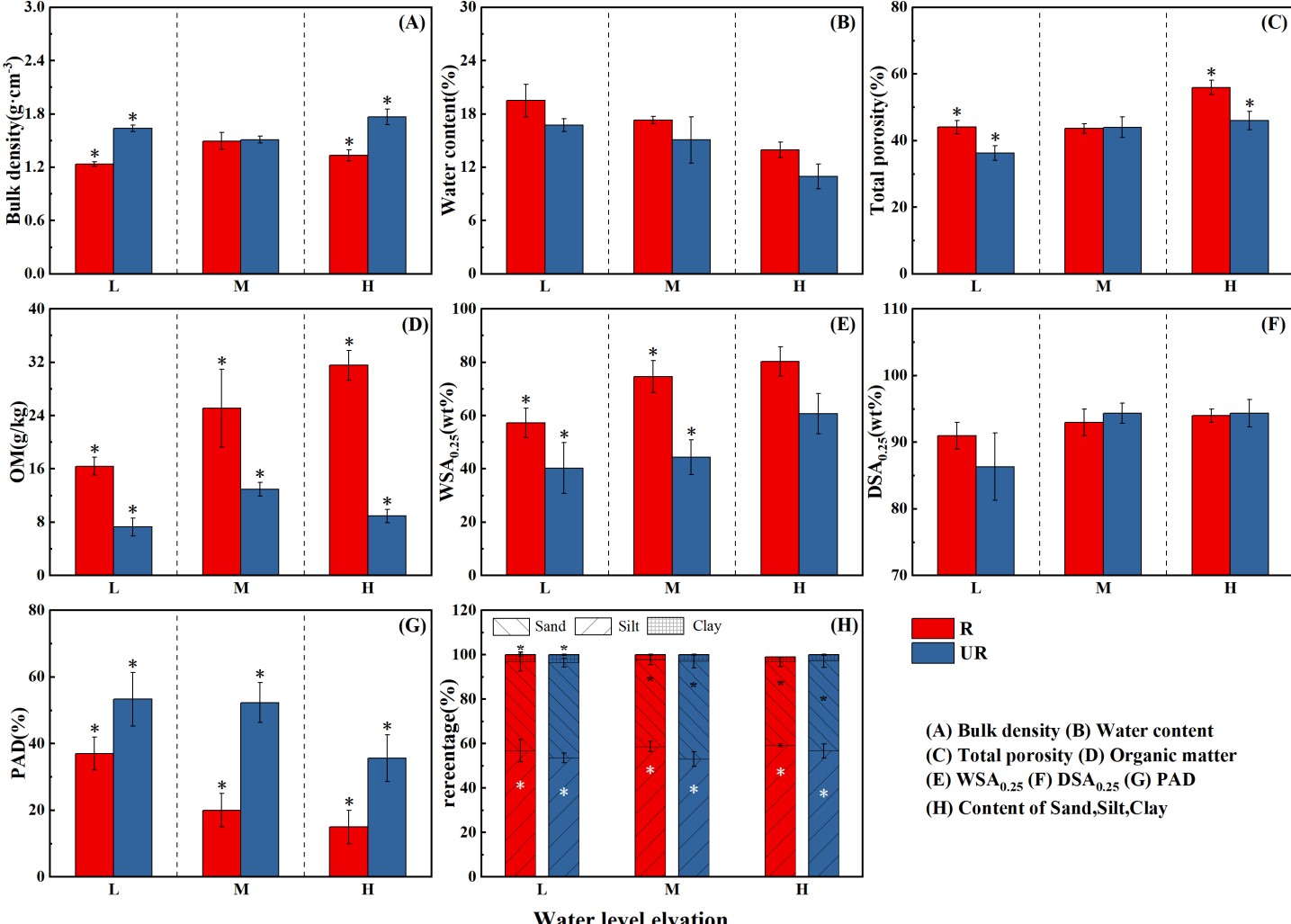

**Fig 3. Changes in soil physicochemical properties at different water level elevations.** R: Root soil, UR: Un-rooted soil *indicates that there are significant differences (p<0.05) in the root soil and un-rooted soils at the same elevation.

The soil moisture content ranged from 10.95% to 19.50%, as shown in Fig 3-B. With increasing elevation, the soil moisture content within the intertidal zone tends to decrease. Compared with that at low elevations, the soil moisture content at middle and high elevations decreased by 11.23% and 19.47%, respectively, in the rhizosphere, whereas the decreases were 10.03% and 27.34%, respectively, within the un-rooted bulk soil. Additionally, the soil moisture content in the rhizosphere soil was greater than that in the un-rooted bulk soil, with increases of 16.42%, 14.86%, and 27.31% at low, middle and high elevations, respectively. There was no significant difference in moisture content between the root zone soil and the un-rooted zone soil at low, middle, and high elevations (P > 0.05).

The measured total soil porosity ranged from 36.29% to 55.94%, as shown in Fig 3-C. The total soil porosity tends to increase with increasing elevation. In the rhizosphere, it increased by 0.09% and 26.96% at middle and high elevations, respectively. In the un-rooted bulk soils, the porosity increased by 20.25% and 5.43% at middle and high elevations, respectively, compared with that at low elevations. At various elevations, the total porosity of the root zone soils is consistently greater than that of the un-rooted zone soils, increasing by 21.30%, 0.96%, and 21.58%, respectively, from low to high elevations. In the middle–high elevations, there was no significant difference between the root zone soils and the un-rooted zone soils (P < 0.05), whereas significant differences existed among the other elevations (P < 0.05).

The measured soil sand content ranged from 53.00% to 59.26%. Across different elevations within the intertidal zone, the sand content of soils tends to increase from low to high elevations. The sand content in the root zone soil was significantly greater than that in the un-rooted zone soil in the low water level area (P < 0.05). On the other hand, the silt and clay contents in the root zone soils are both lower than those in the un-rooted zone soils, showing a decreasing trend with increasing elevation.

The measured soil organic matter content ranged from 7.28 to 31.93 g/kg. With increasing elevation, the soil organic matter content gradually increased. In the root zone soils, there was a sequential increase of 53.08% and 24.87% from low to high elevations. In the un-rooted zone soils, there was an increase of 78.02% from low to middle elevations, followed by a decrease of 31.17% from middle to high elevations. At high elevations, the organic matter content in the root zone soil was significantly greater than that in the un-rooted zone soil (P < 0.05). From low elevations to high elevations, the organic matter content in the root zone soil was 2.25, 1.94, and 3.53 times greater than that in the un-rooted zone soil.

## Soil anti-erosion analysis

**Distribution of soil water-stable aggregates.** At various altitudinal levels, as shown in Fig 4, the proportion of water-stable soil aggregates <0. 25 mm was the highest, ranging from 20.07% to 59.75%. Among the different altitudinal levels, the <0. 25 mm aggregates were predominant in the soil of the un-rooted zones at lower elevations, constituting 59.75% of the total. For different sizes of > 0.25 mm soil aggregates (large aggregates), the proportion of > 5 mm aggregates fluctuated significantly with increasing elevation, ranging between 3.19% and 25.96%. Among these, the upper root zone soil in the decline zone had the highest proportion, accounting for 25.96%. The proportion order of > 5 mm soil aggregates at different water levels was as follows: H-R > H-UR > M-R > M-UR > L-R > L-UR. At various elevations, the proportion of >5 mm soil aggregates in the root zone soil was greater than that in the un-rooted zone soil. However, the percentage variations of the 2–5 mm, 1–2 mm, 0.5–1 mm, and 0. 25–0.5 mm aggregates are relatively stable. With increasing elevation, the soil $WSA_{0.25}$ content in the decline zone gradually increased. In the root zone soil, the content increases sequentially by 30.25% and 7.13% with increasing elevation. In the un-rooted zone soil, the content increases sequentially by 10.83% and 36.86% with elevation. At low and high elevations, the content of water-stable aggregates in rhizosphere soil was significantly greater than that in un-rooted zone soil (P < 0.05). Specifically, at the same elevation, the content of water-stable aggregates in rhizosphere soil was 1.43, 1.68, and 1.32 times greater than that in un-rooted zone soil, respectively, from low to high.

Fig 5 shows that the measured soil mean weight diameter (MWD) ranges from 0.62 to 2.70 mm. With increasing elevation, the MWD of the decline zone soil gradually increases. Specifically, the mean weight diameter (MWD) of rhizosphere

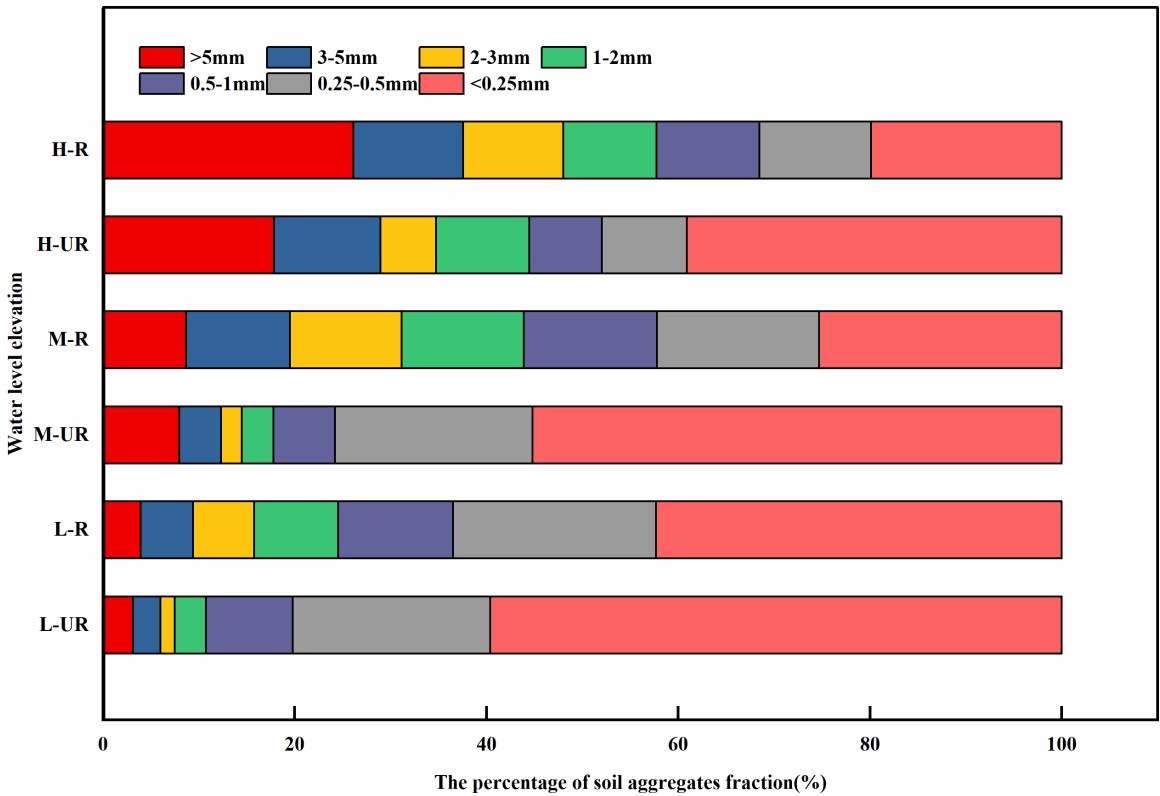

**Fig 4. The proportion of each particle size of water-stable aggregates at different water levels and elevations UR: Un-rooted zone soil, R: root zone soil. L-Low, M-Mid, H-High.**

soil increased by 70.41% and 61.68%, respectively, with elevation, whereas the MWD of non-un-rooted zone soil increased by 58.06% and 107.14%, respectively, with elevation. Overall, the MWD of rhizosphere soil was greater than that of un-rooted zone soil. At middle and high elevations, the MWD of rhizosphere soil was significantly greater than that of un-rooted zone soil ($P < 0.05$). From low to high elevations, the MWD of rhizosphere soil was 1.58, 1.70, and 1.33 times greater than that of un-rooted zone soil.

The measured soil GWD ranged from 0.33 to 1.27 mm. With increasing elevation, the GMD of the decline zone soil gradually increases. Specifically, in the root zone soil, the GMD increases sequentially by 85.71% and 61.54% with elevation. In the un-rooted zone soil, the GMD increases sequentially by 22.22% and 124.24% with elevation. At middle and high elevations, the GMD of rhizosphere soil was significantly greater than that of un-rooted zone soil ($P < 0.05$). Specifically, at the same elevation, the GMD of the rhizosphere soil was 1.56, 2.36, and 1.70 times greater than that of the un-rooted zone soil from low to high, respectively.

**The erodibility of the fluctuation zone.** Fig 6 revealed that the erodibility of the root zone soil decreases from low to high elevations. Specifically, at different elevations, the erodibility decreases from low to middle by 55.26%, and from middle to high by 58.82%. At different elevations, the erodibility of the un-rooted zone soil also tends to decrease from low to high. Specifically, at different elevations, the erodibility decreases from low to middle by 19.35%, and from middle to high, it decreases by 64.00%. Overall, for both the root zone soil and the un-rooted zone soil, the erodibility of the soil decreases with increasing elevation. The erodibility of root–soil complexes is often lower than that of root–soil complexes. At low and middle elevations, the erodibility of the root zone soil is significantly greater than that of the un-rooted zone soil

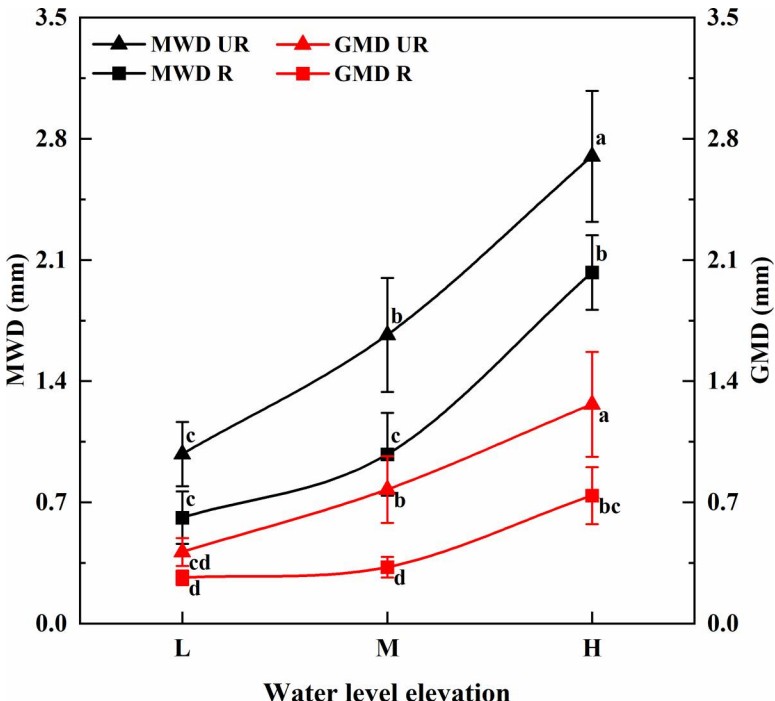

**Fig 5. MWD and GMD values of soil water-stable aggregates in different elevation subsidence zones.** Different uppercase and lowercase letters indicate that there are significant differences at the 0.05 level in different elevations.

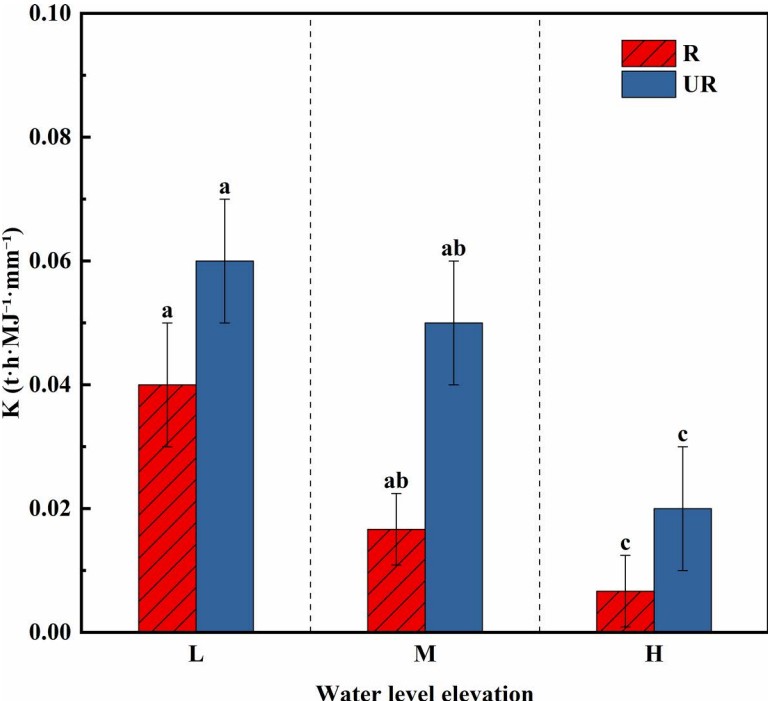

**Fig 6. Erodibility factor (K value) of different elevations in the subsidence zone of Guanyinyan Reservoir area.** Different lowercase letters indicate that there are significant differences at the 0.05 level in different elevations.

(P<0.05). Specifically, at the same elevation, the erodibility of the root zone soil at lower elevations is 0.61 times greater than that of the un-rooted zone soil. At middle--to-high elevations, the erodibility of the root zone soil is 0.34 times greater than that of the un-rooted zone soil. In the upper part of the colluvial zone, the erodibility of the root zone soil is 0.39 times greater than that of the un-rooted zone soil. Thus, vegetation roots play a positive role in soil erosion resistance, especially in the central and lower regions of colluvial zones, which are often subjected to waterlogging. The roots have a more pronounced effect on enhancing the resistance of the soil to erosion in these areas.

**Soil shear strength of fluctuation zone.** According to Fig 7, the measured soil shear strength ranges from 12.33 to 119.33 kPa. With increasing elevation, the shear strength of the colluvial zone soil gradually increases. At low and middle elevations, the shear strength of the root zone soil was significantly greater than that of the nonroot zone soil (P<0.05), whereas at high elevation, the shear strength of the un-rooted zone soil was significantly greater than that of the root zone soil (P<0.05). The percentage of un-rooted zone soil increases by 673.15% and 25.17%, and the percentage of root zone soil increases by 317.12% from low to middle and then decreases by 24.56% from middle to high. At low elevations, the shear strength of the root zone soil is 2.22 times greater than that of the un-rooted zone soil; at middle elevations, it is 1.20 times greater; and at high elevations, it is 0.72 times greater.

## Comprehensive evaluation of soil erosion resistance at different elevations in the colluvial zone

Fig 8 shows that the soil CSEI of the water-level-fluctuating zone in the GYY reservoir area varies between 0.082 and 0.942 with increasing water level elevation. At different elevations, the comprehensive index of soil erosion was as follows: high (root zone soil) 0.082<middle (root zone soil) 0.330<high (un-rooted zone soil) 0.377<middle (un-rooted zone soil) 0.649<low (root zone soil) 0.668<low (un-rooted zone soil) 0.942. At the same water level elevation, with decreasing flooding time, the CSEI of the un-rooted zone soil in the hydro fluctuation belt increased by 41.03%, 96. 91% and 353.

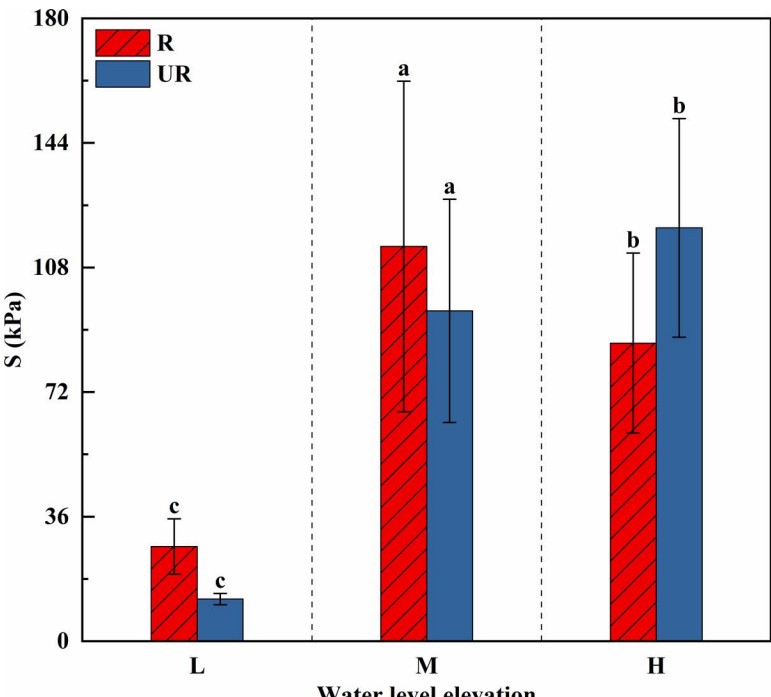

**Fig 7. Shear Strength(S) of different elevations in the subsidence zone of Guanyinyan Reservoir area.** Different lowercase letters indicate that there are significant differences at the 0.05 level in different elevations.

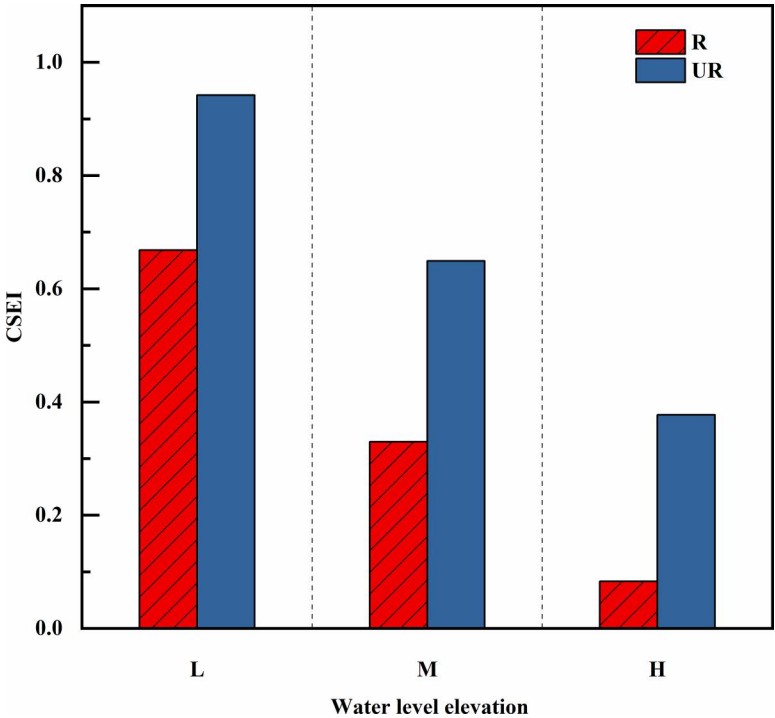

**Fig 8. Variation of CSEI with water level elevation.**

13%, respectively, compared with that of the root zone soil. The higher the elevation is, the shorter the flooding time is, the better the soil erosion resistance, and the root zone soil is often better than that of un-rooted zone soil. This finding is essentially consistent with the results of the previous analysis, indicating that the model can accurately evaluate the soil anti-erodibility of the riparian zone at different elevations in the GYY Reservoir area of the Jinsha River Basin.

## Discussions

### The impact of water level changes on soil physicochemical properties and plant cover

The physical and chemical properties of the soil in the water level fluctuation zone of the Guanyinyan Reservoir area of the Jinsha River significantly differ because of periodic water level changes and alternating wet and dry conditions. These variations may be related to the water level fluctuation cycle, duration of submersion, vegetation growth status, and slope at different elevations. The research results indicate that as elevation increases, the soil moisture content in the water level fluctuation zone tends to decrease. Specifically, in low-elevation areas, owing to the periodic fluctuations in the water level, the soil maintains a relatively high moisture content over a relatively long period. In contrast, in high-elevation areas, the soil in the water level fluctuation zone has a relatively low moisture content. This phenomenon is attributed to the influence of the duration of flooding in the water level fluctuation zone, with soil moisture gradually decreasing from low to high elevations.

The difference in moisture content between soil with roots and soil without roots is particularly significant. Owing to the water retention and interception functions of plant roots, the soil moisture content in the root zone decreases faster with increasing elevation. These findings indicate that the water retention function of plant roots in the water level fluctuation zone plays a crucial role in soil moisture retention at different elevations [23]. In addition, relatively high plant species richness can significantly increase the drought resistance of vegetation communities, thereby improving the ability of plant communities in high-elevation areas to recover from drought after floodwaters recede [24].

The study also revealed that the total porosity of the soil tended to increase, which is consistent with the findings of Duong Thi and Do Minh [25]. Soils that are submerged for extended periods experience a reduction in porosity and an increase in bulk density (BD) due to changes in hydrostatic pressure, which further exacerbates soil degradation. The hydrostatic pressure increases with increasing water depth, intensifying this degradation. As the water depth increases, the hydrostatic pressure increases, causing soils in fluctuating water level zones to undergo more significant structural changes under prolonged flooding conditions. At the same water level elevation, the capacity of root-zone soils is significantly lower than that of un-rooted zone soils, which is consistent with the findings of Wang et al. [26]. The impact of vegetation on soil structure occurs mainly through root stabilization and the accumulation of litter, which improve the soil. As vegetation growth intensifies, the amount of litter increases, and the root system becomes more developed, enhancing soil stabilization and improvement effects, as well as increasing the resistance of the soil to erosion [27].

With frequent fluctuations in water levels, fine soil particles and nutrients are continuously washed away and lost, leading to significant changes in soil structure. Research shows that the more water level cycles the soil undergoes due to these fluctuations, and the longer the soil is subjected to erosion and scouring, the greater the loss of fine particles and nutrients from the soil [28]. In addition, the composition of sediment particles gradually becomes coarser with increasing elevation, which is consistent with the findings of Li et al. [29]. As the elevation increases, the proportion of sediments in riverbank deposits increases, indicating that during low water periods, when water levels are high and the supply of suspended sediments is low, local riverbank erosion intensifies, leading to a continuous increase in sediment yield. During wave erosion, coarse particles may preferentially settle in relatively high riverbank areas, while fine particles may continue to be transported by flow.

The soil organic matter content plays a key role in the formation and stability of aggregates and therefore has an important effect on soil erosion [30]. This study revealed that the soil organic matter content in the upper part of the water level fluctuation zone was relatively high. The flooding duration in the upper part of the water level fluctuation zone is relatively short, providing favorable conditions for plant growth. Compared with that in the lower part, the vegetation coverage in the upper part of the water level fluctuation zone was significantly greater, resulting in relatively higher soil organic matter content in the upper part of the soil [31]. In addition, after the water level decreased, plants at different elevations were able to recover [32], and the aboveground and belowground biomasses at most elevations increased significantly over time., and the aboveground and belowground biomasses at most elevations increased significantly over time. This may be because root-free soil lacks surface vegetation cover. Especially during heavy summer rainfall, nutrients in the topsoil are easily lost through runoff, whereas the rhizosphere soil not only prevents nutrient loss but also increases the organic matter content. Plant root exudates provide ample organic matter for the formation of soil aggregates, thereby promoting the stability of soil aggregates [33]. In the study area, the plants are mainly herbaceous, and their root systems have a certain fibrous function, which further enhances soil stability [34]. Fine roots, especially those associated with fungal hyphae, have a significant promoting effect on microbial activity and help facilitate the formation of soil aggregates by combining humus and other microbial byproducts [35].

## The influence of basic physical and chemical properties of soil and plant cover on soil erosion

Soil erosion resistance refers to the ability of a soil to withstand erosion caused by water flow, raindrops, and other external forces. It is typically determined by multiple factors, including the mechanical composition, aggregate stability, and organic matter content of the soil [36]. Through correlation analysis (Table 8), this study revealed that the soil erodibility factor K in the water level fluctuation zone of the Guanyinyan Reservoir area is significantly correlated with multiple basic physicochemical properties of the soil. Specifically, the soil total porosity, sand content, organic matter content, mean weight diameter (MWD) of the aggregates, geometric mean diameter (GMD), water-stable aggregates larger than 0.25 mm (WSA$_{0.25}$), and dry-sieved aggregates larger than 0.25 mm (DSA$_{0.25}$) were significantly negatively correlated with the erodibility factor K ($P < 0.01$). In contrast, the silt content, clay content, and the percentage of aggregate destruction

**Table 8. Correlations of soil properties across elevation zones.**

|   | BD | TP | Water content | Sand | Silt | Clay | OM |
|---|---|---|---|---|---|---|---|
| K | 0.124 | −0.77** | 0.294 | −0.628** | 0.603** | 0.719** | −0.654** |
| S | 0.307 | 0.353 | −0.569* | 0.104 | −0.035 | −0.501* | 0.189 |
|   | MWD | GMD | WSA$_{0.25}$ | DSA$_{0.25}$ | PAD | K | S |
| K | −0.931** | −0.909** | −0.960** | −0.684** | 0.944** | 1 | −0.508* |
| S | 0.481* | 0.351 | 0.372 | 0.556* | −0.333 | −0.508* | 1 |

"*" and "**" mean significant correlation at 0.05 and 0.01 levels, respectively. Concentrate: BD: bulk density; TP: total porosity; K: soil erodibility factor; S.: shear strength; OM: organic matter; MWD: average weight diameter; GMD: geometric mean diameter; WSA$_{0.25}$: water-stable aggregates larger than 0.25 mm; DSA$_{0.25}$: dry-sieved aggregates larger than 0.25 mm; PAD: percentage of aggregate destruction for aggregates larger than 0.25 mm.

for aggregates larger than 0.25 mm (PAD) were positively correlated with the erodibility factor K (P < 0.01). These results indicate that higher organic matter content and more stable aggregates in the soil play important roles in enhancing soil resistance to erosion.

Aggregate stability (MWD and GMD) is a key indicator for measuring soil erosion resistance and is typically influenced by a combination of factors such as soil texture, moisture, and organic matter content [37]. Research shows that water-stable aggregates are typically cemented together by clay particles and organic matter, and these aggregates play a crucial role in resisting water erosion [38]. This study revealed that the stability of aggregates in the water level fluctuation zone is closely related to water level changes and the effects of long-term flooding. As the water level increased, the organic matter (OM) content in the soil increased, as did the levels of water-stable aggregates larger than 0.25 mm (WSA$_{0.25}$) and Dry-sieved aggregates larger than 0.25 mm (DSA$_{0.25}$), which increased the stability of the soil aggregates and reduced soil erodibility. On the other hand, the sand content is negatively correlated with the erodibility factor K, which may be related to the larger size of the sand particles and their tendency to be easily carried away by water flow. Therefore, soils with higher sand contents have surface particles that are more easily washed away by water flow, increasing the soil erodibility. In contrast, higher contents of clay and silt increase the cohesiveness of the soil, slowing water erosion and thereby enhancing the resistance of the soil to erosion [39].

Vegetation cover has a significant effect on soil erosion resistance, primarily through the "anchoring" effect of its root system on the soil. Compared with those in the upper area, the vegetation coverage in the lower part of the water level fluctuation zone is lower, and the soil is submerged by the water level for a longer period, resulting in lower resistance to erosion. Prolonged flooding causes the organic matter in the soil to be submerged and washed away, which reduces the stability of soil aggregates and increases the likelihood of erosion [40]. Compared with that in the upper area, the vegetation coverage in the lower part of the water level fluctuation zone is lower, and the soil is submerged by water for a longer period of time, resulting in lower soil resistance to erosion [41]. In addition, plant roots further reduce soil erosion by slowing the flow of water [42]. The vegetation in the water level fluctuation zone also increases the organic matter content in the topsoil through the deposition of plant residues, further increasing the resistance of the soil to erosion.

Soil shear strength is an important indicator for assessing the resistance of soil to erosion. This study revealed that the soil shear strength is significantly negatively correlated with the soil moisture content, clay content, and erodibility factor K, whereas it is significantly positively correlated with the mean weight diameter (MWD) of the aggregates and the dry-sieved aggregates larger than 0.25 mm (DSA$_{0.25}$) (P < 0.05). These results indicate that the greater the shear strength of a soil is, the greater its ability to resist erosion caused by water flow and raindrops, and the lower the risk of soil erosion. In the study area, soils with higher organic matter content and more stable aggregates tend to have greater shear strength, effectively preventing erosion by water flow [43]. There is also a potential negative correlation between root density and shear strength, which may be related to soil softening caused by increased soil moisture. When soil moisture increases, the cohesion between soil particles weakens, causing the originally compact soil structure to become loose, leading to

a decrease in overall stability and a corresponding reduction in shear strength [44]. The soil-stabilizing effect of vegetation root systems plays an important role in enhancing soil shear strength. Roots increase soil cohesion through physical anchoring and chemical bonding, thereby improving its shear strength. In particular, in the upper part of the fluctuating water level zone, due to greater vegetation coverage and shorter flooding duration, the soil maintains relatively high shear strength, reducing the occurrence of soil erosion [45].

Through Redundancy analysis (RDA), the combined effects of basic soil physicochemical properties and plant cover on soil erosion were further revealed. The RDA results indicate that the soil texture and aggregate stability have significant impacts on the soil erosion resistance and shear strength, with the soil moisture content being a key factor influencing the soil shear strength. In areas with higher moisture contents, the soil shear strength is lower, making the soil more susceptible to erosion; conversely, in areas with lower moisture contents and more stable aggregates, the soil erosion resistance and shear strength are greater. The study also revealed that the soil erodibility factor K is negatively correlated with the soil organic matter content and aggregate stability, indicating that organic matter and aggregate stability are crucial for soil erosion resistance. Vegetation cover significantly enhances soil erosion resistance by increasing the soil organic matter content and improving the soil structure, further confirming the important role of vegetation in soil erosion control.

### Effect of Comprehensive Soil Erosion Index (CSEI)

In this study, the calculated CSEI increases with the decrease of water level elevation (Fig 8). These results indicate that the soil is more likely to erode with the increase of submergence time and wetting-drying cycle frequency. The variation of CSEI across elevations was mainly caused by changes in total porosity, organic matter content, soil mechanical composition, and aggregates (Fig 3 and Fig 4). These changes resulted from variations in submergence intensity, duration, and wet-dry cycle frequency. The results of linear regression analysis showed that CSEI was negatively correlated with TP, Sand, OM, MWD, GMD, $WSA_{0.25}$ and $DSA_{0.25}$, and positively correlated with Silt, Clay, PAD and K (Fig 9). The relationship between CSEI and other indicators is shown in Fig 10. Cui and Tang et al [46] found that the lower part of the water-level-fluctuating zone is often subject to the most serious soil erosion caused by flooding and dry-wet cycles. In general, the low-elevation areas in winter are completely submerged, and the hydrostatic pressure causes the soil pores to be completely filled with water, resulting in the expansion and decomposition of soil.

With the approaching of summer, the water level drops rapidly, and the soil in the low elevation area of the hydro-fluctuation belt is completely exposed. Rainstorms always occur in the season of TGR [47]. The splash of raindrops will lead to the decomposition of soil aggregates and reduce soil porosity, thus causing changes in soil structure. Further analysis showed that long-term submergence and frequent alternation of wetting and drying led to significant deterioration of soil structure at low elevations. With the increase of elevation, the inundation intensity decreases and the soil degradation gradually disappears. Jiang and Shi et al [48] found that the soil in the hydro-fluctuation belt was softened and muddy by water during the rise of water level. At the same time, the soil is eroded by rainfall and soil erosion is serious. In addition, vegetation roots play an important role in resisting soil erosion and stabilizing reservoir banks. Plants establish a dense root network in their underground parts, which greatly reduces soil porosity and binds soil aggregates together [49]. Moreover, plant root exudates provide the necessary organic matter for the formation of soil aggregates and promote the formation of aggregates. The plants in the study area are mainly herbaceous, and the roots play a fiber role to a certain extent, which improves soil stability. Water level fluctuations affect the distribution of plants, and promote the agglomeration of large aggregate particles through root and soil consolidation, thereby improving the stability of soil aggregates.

### Conclusion

1) In the Jinping Guanyinyan Reservoir area, the erosion zone is influenced by the periodic rise and fall of the reservoir water level. The erosion zone in the reservoir area often undergoes a regular process of alternating wetting and drying, leading to changes in the physicochemical properties of the soil in this zone. At different water level elevations, the soil

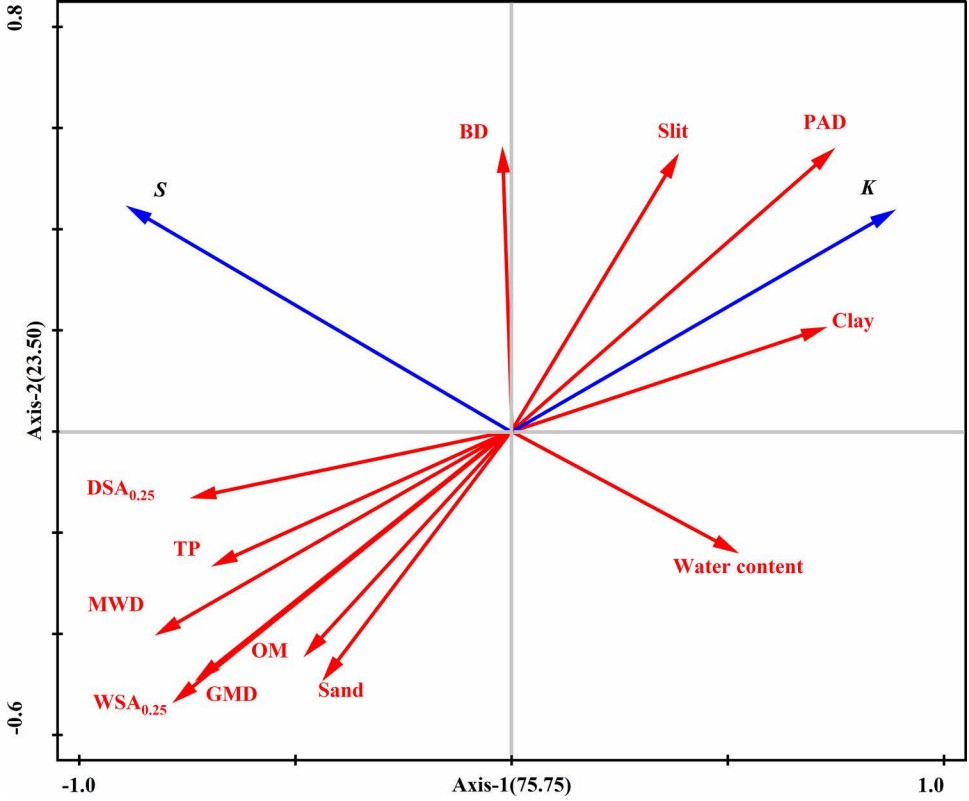

**Fig 9. Redundancy analysis (RDA) of the relationship between soil physicochemical properties and soil erodibility factor K and shear strength in the riparian zone.** BD: Bulk density; TP: total porosity; OM: organic matter; K: soil erodibility factor; S: shear strength; $WSA_{0.25}$: water-stable aggregates larger than 0.25 mm; $DSA_{0.25}$: dry-sieved aggregates larger than 0.25 mm; PAD: percentage of aggregate destruction for aggregates larger than 0.25 mm; MWD: mean weight diameter; GMD: geometric mean diameter.

erodibility factor K tends to decrease with increasing water level elevation. In the root zone, the order is as follows: high ($0.007 \; t \cdot h \cdot MJ^{-1} \cdot mm^{-1}$) < middle ($0.017 \; t \cdot h \cdot MJ^{-1} \cdot mm^{-1}$) < low ($0.038 \; t \cdot h \cdot MJ^{-1} \cdot mm^{-1}$); in the un-rooted zone, the order is as follows: high ($0.018 \; t \cdot h \cdot MJ^{-1} \cdot mm^{-1}$) < middle ($0.05 \; t \cdot h \cdot MJ^{-1} \cdot mm^{-1}$) < low ($0.062 \; t \cdot h \cdot MJ^{-1} \cdot mm^{-1}$). At the same water level elevation, the un-rooted zone soil is more susceptible to erosion than the root zone soil is, with increases of 1.63 times, 2.94 times, and 2.57 times from low to high.

2) Redundancy analysis (RDA) and Pearson correlation analysis revealed significant negative correlations ($P < 0.01$) between the soil erodibility factor K in the Jinping Gaunyinyan Reservoir erosion zone and total porosity, sand content, organic matter, mean weight diameter (MWD), geometric mean diameter (GMD), water-stable aggregates larger than 0.25 mm ($WSA_{0.25}$), and dry-sieved aggregates larger than 0.25 mm ($DSA_{0.25}$), with correlation coefficients of −0.77, −0.63, −0.65, −0.93, −0.91, −0.96, and −0.68, respectively. There were also significant positive correlations ($P < 0.01$) with the silt content, clay content, and percentage of aggregate destruction for aggregates larger than 0.25 mm (PAD), with correlation coefficients of 0.60, 0.72, and 0.94, respectively. Among these factors, the MWD, GMD, and $WSA_{0.25}$ played crucial roles in soil erosion resistance. The higher their content is, the stronger the erosion resistance of the soil. The soil shear strength was significantly negatively correlated with the moisture content, clay content, and soil erodibility factor K, with correlation coefficients of −0.57, −0.50, and −0.51, respectively. Significant positive correlations ($P < 0.05$) were found with the MWD and $DSA_{0.25}$, with correlation coefficients of 0.48 and 0.56, respectively. RDA redundancy analysis indicated that various indicators had a shallower impact on the soil shear strength than they

**Fig 10. Relationship between CSEI and TP, Sand, Silt, Clay, OM, MWD, GMD, WSA$_{0.25}$, DSA$_{0.25}$, PAD, and K.**

did on the soil erodibility factor K. The soil moisture content was identified as a key factor influencing the soil shear strength. High shear strength in the soil corresponded to strong resistance against surface runoff and raindrop splash erosion.

3) Fourteen erosion resistance indicators of the colluvial zone soil in the Guanyinyan Reservoir area were selected, and based on principal component analysis (PCA), they were optimized into three main components: Y1 [TP (X2), MWD (X8), GMD (X9), $WSA_{0.25}$ (X10), PAD (X12), K (X13)] with 6 factors; Y2 [Water content (X2), Shear strength (X14)] with 2 factors; and Y3 [BD (X2), Sand (X4), Silt (X5), Clay (X6), OM (X7), $DSA_{0.25}$ (X11) content] with 6 factors. The comprehensive evaluation model for soil erosion resistance is then formulated as $Y = 0.702Y1 + 0.214Y2 + 0.084Y3$.

4) The Comprehensive Soil Erosion Index (CSEI) in the Jinping Guanyinyan Reservoir erosion zone varies between 0.082 and 0.942 with changes in water level elevation. For different elevations, the comprehensive soil erosion index is as follows: high (root zone soil) 0.082 < middle (root zone soil) 0.330 < high (un-rooted zone soil) 0.377 < middle (un-rooted zone soil) 0.649 < low (root zone soil) 0.668 < low (un-rooted zone soil) 0.942. At the same water level elevation, with decreasing flooding time, the CSEI of the un-rooted zone soil in the erosion zone increased by 41.03%, 96.91%, and 353.13% compared with that of the root zone soil. Vegetation restoration is considered an effective measure to prevent soil erosion in erosion zones. However, the current impact mechanism of plant communities on the susceptibility of the erosion zone to soil erosion requires further research.

## Supporting information

**S1 File. Supplemental tables, figures, and detailed data.**
(ZIP)

## Acknowledgments

Thanks to Jiazheng Mo and Zefan Huang for their supports. The authors are also grateful to the anonymous reviewers for their helpful comments and advice.

## Author contributions

**Conceptualization:** Xinghan Niu, Zexi Song.

**Formal analysis:** Henglin Xiao.

**Investigation:** Pengcheng Wang.

**Methodology:** Pengcheng Wang, Henglin Xiao.

**Supervision:** Henglin Xiao, Gaoliang Tao.

**Writing – original draft:** Pengcheng Wang, Zexi Song.

**Writing – review & editing:** Gaoliang Tao.

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
