## [Decision Letter · Decision Letter 0]

29 Jun 2025

Dear Dr. 宋,

Thank you for submitting your manuscript to PLOS ONE. After careful consideration, we feel that it has merit but does not fully meet PLOS ONE’s publication criteria as it currently stands. Therefore, we invite you to submit a revised version of the manuscript that addresses the points raised during the review process.

We look forward to receiving your revised manuscript.

Kind regards,

Somayeh Soltani-Gerdefaramarzi, Ph. D.

Academic Editor

PLOS ONE

**Journal Requirements:**

1. When submitting your revision, we need you to address these additional requirements. Please ensure that your manuscript meets PLOS ONE's style requirements, including those for file naming. The PLOS ONE style templates can be found at https://journals.plos.org/plosone/s/file?id=wjVg/PLOSOne_formatting_sample_main_body.pdf and https://journals.plos.org/plosone/s/file?id=ba62/PLOSOne_formatting_sample_title_authors_affiliations.pdf 2. In your Methods section, please provide additional information regarding the permits you obtained for the work. Please ensure you have included the full name of the authority that approved the field site access and, if no permits were required, a brief statement explaining why. 3. We note that the grant information you provided in the ‘Funding Information’ and ‘Financial Disclosure’ sections do not match.  When you resubmit, please ensure that you provide the correct grant numbers for the awards you received for your study in the ‘Funding Information’ section. 4. Thank you for stating the following in the Acknowledgments Section of your manuscript: This research was supported by the National Natural Science Foundation of China (No. 42307256), the Joint Funds of the Nature Science Foundation of Hubei Province (No. 2022CFD172), the Joint Funds of the National Nature Science Foundation of China (U22A20232), the Open Project Funding of Hubei Key Laboratory of Environmental Geotechnology and Ecological Remediation for Lake & River (HJKFYB202405) and the Innovation Demonstration Base of Ecological Environment Geotechnical, Ecological Restoration of Rivers and Lakes (2020EJB004). We note that you have provided funding information that is not currently declared in your Funding Statement. However, funding information should not appear in the Acknowledgments section or other areas of your manuscript. We will only publish funding information present in the Funding Statement section of the online submission form. Please remove any funding-related text from the manuscript and let us know how you would like to update your Funding Statement. Currently, your Funding Statement reads as follows: The author(s) received no specific funding for this work.  Please include your amended statements within your cover letter; we will change the online submission form on your behalf. 5. In the online submission form, you indicated that “The datasets used during this study are available from the corresponding authors upon reasonable request”. All PLOS journals now require all data underlying the findings described in their manuscript to be freely available to other researchers, either a. In a public repository, b. Within the manuscript itself, or c. Uploaded as supplementary information.This policy applies to all data except where public deposition would breach compliance with the protocol approved by your research ethics board. If your data cannot be made publicly available for ethical or legal reasons (e.g., public availability would compromise patient privacy), please explain your reasons on resubmission and your exemption request will be escalated for approval. 6. One of the noted authors is a group or consortium. In addition to naming the author group, please list the individual authors and affiliations within this group in the acknowledgments section of your manuscript. Please also indicate clearly a lead author for this group along with a contact email address. 7. We note that Figure 1 in your submission contain map images which may be copyrighted. All PLOS content is published under the Creative Commons Attribution License (CC BY 4.0), which means that the manuscript, images, and Supporting Information files will be freely available online, and any third party is permitted to access, download, copy, distribute, and use these materials in any way, even commercially, with proper attribution. For these reasons, we cannot publish previously copyrighted maps or satellite images created using proprietary data, such as Google software (Google Maps, Street View, and Earth). For more information, see our copyright guidelines: http://journals.plos.org/plosone/s/licenses-and-copyright. We require you to either present written permission from the copyright holder to publish these figures specifically under the CC BY 4.0 license, or remove the figures from your submission: a. You may seek permission from the original copyright holder of Figure 1 to publish the content specifically under the CC BY 4.0 license.   We recommend that you contact the original copyright holder with the Content Permission Form (http://journals.plos.org/plosone/s/file?id=7c09/content-permission-form.pdf) and the following text:“I request permission for the open-access journal PLOS ONE to publish XXX under the Creative Commons Attribution License (CCAL) CC BY 4.0 (http://creativecommons.org/licenses/by/4.0/). Please be aware that this license allows unrestricted use and distribution, even commercially, by third parties. Please reply and provide explicit written permission to publish XXX under a CC BY license and complete the attached form.” Please upload the completed Content Permission Form or other proof of granted permissions as an "Other" file with your submission. In the figure caption of the copyrighted figure, please include the following text: “Reprinted from [ref] under a CC BY license, with permission from [name of publisher], original copyright [original copyright year].” b. If you are unable to obtain permission from the original copyright holder to publish these figures under the CC BY 4.0 license or if the copyright holder’s requirements are incompatible with the CC BY 4.0 license, please either i) remove the figure or ii) supply a replacement figure that complies with the CC BY 4.0 license. Please check copyright information on all replacement figures and update the figure caption with source information. If applicable, please specify in the figure caption text when a figure is similar but not identical to the original image and is therefore for illustrative purposes only.The following resources for replacing copyrighted map figures may be helpful: USGS National Map Viewer (public domain): http://viewer.nationalmap.gov/viewer/The Gateway to Astronaut Photography of Earth (public domain): http://eol.jsc.nasa.gov/sseop/clickmap/Maps at the CIA (public domain): https://www.cia.gov/library/publications/the-world-factbook/index.html and https://www.cia.gov/library/publications/cia-maps-publications/index.htmlNASA Earth Observatory (public domain): http://earthobservatory.nasa.gov/Landsat:
http://landsat.visibleearth.nasa.gov/USGS EROS (Earth Resources Observatory and Science (EROS) Center) (public domain): http://eros.usgs.gov/#Natural Earth (public domain): http://www.naturalearthdata.com/

Reviewers' comments:

Reviewer's Responses to Questions

**Comments to the Author**

1. Is the manuscript technically sound, and do the data support the conclusions?

Reviewer #1: Yes

Reviewer #2: Yes

2. Has the statistical analysis been performed appropriately and rigorously?

Reviewer #1: Yes

Reviewer #2: Yes

3. Have the authors made all data underlying the findings in their manuscript fully available?

Reviewer #1: Yes

Reviewer #2: Yes

4. Is the manuscript presented in an intelligible fashion and written in standard English?

Reviewer #1: Yes

Reviewer #2: Yes

**Reviewer #1:** This manuscript presents a comprehensive and well-structured study on the evaluation of soil erosion resistance in the fluctuation zones of a major reservoir in Southwest China. The authors combine detailed physicochemical analyses, soil aggregate stability assessments, and statistical modeling, including Principal Component Analysis and Redundancy Analysis, to develop a soil erosion resistance evaluation model. The focus on rooted versus un-rooted zones and variation across elevation gradients offers valuable insights into soil stability dynamics in reservoir-affected riparian systems.

The research addresses an important ecological and hydrological topic, providing data-driven recommendations for managing soil erosion in hydro-fluctuation zones—an area of growing concern given the global increase in reservoir construction. The methods are appropriate, the data are sound, and the interpretations are generally well-supported by the evidence.

However, before acceptance, the following minor revisions are recommended:

Language and Grammar: While the manuscript is largely understandable, it would benefit from careful proofreading to correct occasional grammatical errors and improve clarity in phrasing. Particular attention should be given to sentence structure in the introduction and discussion sections.

Figure and Table Captions: Some figure and table captions lack sufficient detail. For instance, captions should clearly define abbreviations (e.g., WSA0.25, PAD), explain what the reader is observing, and note any statistical significance levels.

Data Availability Statement: The statement currently indicates that data are available on request. It is recommended to deposit the dataset in a public repository in accordance with PLOS ONE’s data availability policy to enhance transparency and reproducibility.

References Formatting: A few citations in the text appear incomplete or inconsistent with PLOS ONE formatting guidelines. Please ensure that all references are updated and uniformly formatted.

Minor Clarifications:

The use of overlapping abbreviations (e.g., "TP" for total porosity and "TP" elsewhere) should be reviewed to avoid confusion.

Consider briefly justifying the selection of indicators used in PCA in the Methods section to strengthen the rationale for the model structure.

In conclusion, this is a valuable contribution to soil science and watershed management. With minor editorial improvements, it will be well-suited for publication in PLOS ONE.

**Reviewer #2:** I. Structural Issues

• Abstract Section

o Issue: The description of results in the Abstract (e.g., specific numerical ranges and comparisons of the CSEI index) is overly detailed, exceeding the requirement for conciseness. This risks obscuring the key points.

o Suggested Revision: Simplify specific data points; focus on core conclusions (e.g., "CSEI decreases with increasing elevation"). Emphasize key trends rather than precise numerical values.

• Introduction Section

o Issue: The discussion on the role of plant roots (e.g., "root traits may mediate the impact of plant diversity") lacks a tight logical connection to the subsequent research objectives. It fails to explicitly link root systems with the interactive effects of water level fluctuations.

o Suggested Revision: Supplement the explanation with the unique mechanisms of root systems in water fluctuation zones, such as root adaptations to wet-dry cycles and their regulatory role on soil anti-erodibility.

• Discussion Section

o Issue: There is content overlap between Section 4.1 (Impact of Water Level Changes on Soil Physicochemical Properties) and Section 4.2 (Impact of Soil Properties on Erosion). For example, the statement "Soil organic matter content increases with elevation" appears in both sections, creating logical redundancy.

o Suggested Revision: Integrate overlapping content. Reorganize paragraphs according to the logical chain: "Water Level Fluctuations → Changes in Soil Properties → Anti-Erodibility Response".

• Conclusion Section

o Issue: In Conclusion 1, the description of "root zone soil K value variation with elevation" (e.g., "High (0.007) < Medium (0.017) < Low (0.038)") omits units (e.g., "t·h·MJ⁻¹·mm⁻¹"), violating academic standards.

o Suggested Revision: Add the units for the K value and ensure consistent data presentation format throughout the paper.

II. Academic Quality Issues

• Limitations of Research Methods

o Issue: In Section 2.4 (Data Analysis Methods), Principal Component Analysis (PCA) reduced 14 indicators to 3 principal components, but the rationale for selecting the first 3 components (e.g., eigenvalue >1 criterion) is not provided. Reliability tests for the PCA (e.g., KMO test or Bartlett's test of sphericity) are also missing.

o Suggested Revision: Supplement the results of PCA suitability tests (KMO, Bartlett's) and justify the cumulative variance contribution rate (85.143%).

• Insufficient Depth in Data Discussion

o Issue: The analysis of soil shear strength variation with elevation in Section 3.2.3 (e.g., "root zone soil shear strength decreases at high elevation") lacks integration with interacting factors like root density or soil moisture content, making the explanation inadequate.

o Suggested Revision: Correlate with root biomass data (e.g., high elevation vegetation coverage 78%). Analyze that the potential negative correlation between root density and shear strength might be caused by increased soil moisture leading to soil softening.

• Timeliness and Relevance of References

o Issue: Reference [16] (Wischmeier, 1978) is a classic model, but recent advances in the field (e.g., studies on soil erosion in reservoir fluctuation zones post-2020) are not cited. Some references (e.g., [65]) are cited redundantly.

o Suggested Revision: Supplement relevant studies from 2020-2025. Replace redundant citations to enhance the study's currency.

III. Grammatical Issues

• Mixed Chinese-English Formatting and Errors

o Issue: Formula symbols like "DSA 0.25" are inconsistently mixed with Chinese text in the Abstract. Some paragraphs (e.g., paragraph 3 of the Abstract) contain grammatical errors, such as the misspelling "t is sisnificntly positively correlated".

o Suggested Revision: Standardize formula symbol formatting (e.g., use "DSA₀.₂₅"). Correct spelling errors to "it is significantly positively correlated".

• Inconsistent Terminology

o Issue: Terms like "colluvial slope soils" (Abstract) and "colluvial zone soil" (Abstract) are used inconsistently. "Erosion coefficient" and "erodibility factor K" are used interchangeably, causing confusion.

o Suggested Revision: Standardize terms (e.g., use "colluvial zone soil" and "erodibility factor K" consistently). Define full terms upon first appearance.

• Wordy Sentences and Logical Gaps

o Issue: The last paragraph of the Introduction ("Over all the response of soil detachment...") has overly complex sentence structure and grammatical gaps (e.g., missing subject before "are in need of further study").

o Suggested Revision: Break into shorter sentences, e.g.: "However, the quantitative effects of these factors and their dominant influences remain unknown. It is speculated that root traits may mediate...".

IV. Errors in Methods, Knowledge, and Techniques

• Formula Citation and Derivation Issues

o Issue: In Section 2.2, the calculation formulas (1) and (2) for the soil erodibility factor K are cited from Shirazi et al., 1988, but the specific improvements made by Zhang et al., 2008 (e.g., parameter adjustments or applicability extensions) are not explained. Furthermore, "logGMD" in formula (1) lacks specification of the logarithm base (should be natural logarithm ln).

o Suggested Revision: Supplement the specific steps of the improvement method. Correct the formula to "ln GMD" and clearly cite the source.

• Insufficient Rigor in Statistical Analysis

o Issue: In Section 3.1, the statement "root zone soil bulk density decreased by 24.39% compared to non-root zone" lacks indication of statistical significance (e.g., p-value from t-test or ANOVA). Merely describing numerical differences does not meet academic standards.

o Suggested Revision: Supplement significance test results, e.g., "(p < 0.05, t-test)", and add statistical symbols (e.g., asterisks) to Figure 3.

• Ambiguous Definition of Technical Indicators

o Issue: In Section 2.3, the definition of "PAD (X12)" as "aggregate of destruction" is unclear regarding its relationship to "aggregate destruction rate". Additionally, units for "DSA 0.25" and "WSA₀.₂₅" in formula (4) are missing (should be mass percentage).

o Suggested Revision: Clarify that PAD refers to the ">0.25 mm aggregate destruction rate". Supplement the units and physical meaning of all parameters in the formulas.

V. Comprehensive Revision Suggestions

• Structural Optimization: Reorganize the Discussion section logic to avoid repetition. Supplement data units and trend analysis in the Conclusions.

• Academic Rigor: Enhance PCA validation procedures, supplement supporting data (e.g., root biomass), update references.

• Language Standardization: Unify terminology, correct grammatical errors, optimize formula symbol formatting.

• Methodological Refinement: Elaborate on formula improvement details, add statistical significance tests, standardize indicator definitions.

**Do you want your identity to be public for this peer review?** For information about this choice, including consent withdrawal, please see our Privacy Policy

Reviewer #1: **Yes:** Dr. Tancredo Souza

Reviewer #2: **Yes:** Zhenhong Wang

---

## [Author Response · Author response to Decision Letter 1]

24 Sep 2025

Response to the Reviewers

Dear Editors and Reviewers:

Thank you for your letter and for the reviewers’ comments concerning our manuscript entitled “Quantitative evaluation of soil anti-erodibility in the fluctuation zones of rooted soil in a large reservoir, southwest of China” (Manuscript ID: PONE-D-25-21716). Those comments are all valuable and very helpful for revising and improving our paper, as well as the important guiding significance to our research. We have studied comments carefully and made corrections which we hope to meet with approval. Revised portions are marked in the paper. The responds to the reviewer’s comments are as follows:

The information of the corresponding author's institution has changed：

Due to personnel changes, the corresponding author Xiao Henglin's workplace was changed to State Key Laboratory of Precision Blasting, Jianghan University, Wuhan, 430056, China.

Affiliated institution changed to:

1. Hubei Key Laboratory of Environmental Geotechnology and Ecological Remediation for Lake & River, Hubei University of Technology, Wuhan 430068, China.

2. Key Laboratory of Intelligent Health Perception and Ecological Restoration of Rivers and Lakes, Ministry of Education, Hubei University of Technology, Wuhan 430068, China.

3. State Key Laboratory of Precision Blasting, Jianghan University, Wuhan, 430056, China

Changed in the Acknowledgments to:

This research was supported by the National Natural Science Foundation of China (No. 42307256), the Joint Funds of the Nature Science Foundation of Hubei Province (No. 2022CFD172), the Joint Funds of the National Nature Science Foundation of China (U22A20232), the Innovation Research Group Project of the Hubei Provincial Department of Science and Technology (2025AFA020), the Open Project Funding of Hubei Key Laboratory of Environmental Geotechnology and Ecological Remediation for Lake & River (HJKFYB202405), the Innovation Research Team Project of the Hubei Provincial Department of Science (No.T2024006) and the Innovation Demonstration Base of Ecological Environment Geotechnical, Ecological Restoration of Rivers and Lakes (2020EJB004).

The second author of the article has been changed from Zexi Song to Xinghan Niu, and Zexi Song has been moved to the fifth author. Please note this change.

Comments and Suggestions for Authors

Structural Issues

• Abstract Section

Q1: Issue: The description of results in the Abstract (e.g., specific numerical ranges and comparisons of the CSEI index) is overly detailed, exceeding the requirement for conciseness. This risks obscuring the key points.

R1：We sincerely appreciate the reviewer’s insightful feedback regarding the level of detail in the Abstract. We agree that excessive numerical specifics may detract from the core conclusions and compromise conciseness. Accordingly, we have comprehensively revised the Abstract to prioritize clarity and emphasize key trends, as suggested.

In the abstract section, I have removed the specific numerical values and correlation coefficients of CSEI, such as "correlation coefficients of -0.70, -0.5, and -0.52, respectively," and detailed figures like "High (root zone soil) 0.082 < Middle (root zone soil) 0.330 < High (un-rooted zone soil) 0.377 < Medium (un-rooted zone soil) 0.649 < Low (root zone soil) 0.668 < Low (un-rooted zone soil) 0.942." The focus was kept on the key trends and core conclusions to maintain the abstract's conciseness.

•• Introduction Section

Q2: Issue: The discussion on the role of plant roots (e.g., "root traits may mediate the impact of plant diversity") lacks a tight logical connection to the subsequent research objectives. It fails to explicitly link root systems with the interactive effects of water level fluctuations.

R2：We thank the reviewer for this valuable observation. We agree that a clearer mechanistic link between root systems, water-level fluctuations, and soil anti-erodibility was needed to strengthen the logical flow towards our research objectives.

In lines 142-153 of the article, I added content related to how plant roots adapt to environmental changes under alternating dry and wet conditions by altering their morphological structure, thereby affecting erosion resistance, and established a link between the interaction of roots and water level fluctuations.From: "In addition, the effects of plant roots on soil properties also differ with respect to various root characteristics and environmental conditions. Under alternating dry and wet conditions, plant root systems adapt to environmental changes by altering their morphological structure." up to the end of this paragraph is the new content I supplemented. Additionally, I added a paragraph in lines 70-77 of the introduction section to briefly explain what soil erosion is, making the introduction more comprehensive.

• Discussion Section

Q3:Section 4.1 (Impact of Water Level Fluctuations on Soil Physicochemical Properties) and Section 4.2 (Impact of Soil Properties on Erosion) exhibit content overlap.

R3：Thank you for your valuable feedback. As suggested, we have restructured the relevant paragraphs according to the logical framework of "water level fluctuations → changes in soil properties → anti-erosion response." To improve clarity and focus, all content pertaining to anti-erosion effects has been consolidated into Section 4.2, while Section 4.1 now exclusively discusses the influence of water level changes on soil physicochemical properties, without reference to erosion-related aspects. Some overlapping content may remain, and we would be grateful for any further suggestions you may have.

• Conclusion Section

Q4: In Conclusion 1, the description of root zone soil potassium values varying with elevation (e.g., 'high (0.007) < medium (0.017) < low (0.038)') omits the units, violating academic standards.

R4：Thank you for your valuable feedback. We have addressed the points you raised in the corresponding sections. Specifically, the unit of K ("t·h·MJ⁻¹·mm⁻¹") has now been added to Conclusion 1, as well as to formula (2) in Section 2.2.

Academic Quality Issues

Q5: In Section 2.4 (Data Analysis Methods), Principal Component analysis (PCA) reduces 14 indicators to 3 principal components, but does not provide reasons for choosing the first 3 components (for example, eigenvalue >1 criterion). The reliability tests of PCA (such as KMO test or Bartlett sphericity test) are also missing.

R5：Thank you for your valuable feedback. We have incorporated the KMO test and Bartlett’s test of sphericity into the manuscript. Furthermore, based on the scree plot, the first three principal components were retained, as presented below：

KMO Sampling Adequacy Measure 0.647

Bartlett's test of sphericity Approximate chi square 332.68

freedom 66

significance 0

Q6: The analysis of soil shear strength variation with elevation in Section 3.2.3 lacks integration with interacting factors like root density or soil moisture content, making the explanation inadequate.

R6：Thank you for your valuable feedback. Due to the fact that our sampling was focused exclusively on root-related data, we have incorporated corresponding data on changes in plant root systems in relation to water level variations into the concluding segment of the second paragraph in Section 2.2.

Elevation length (cm) surface area (cm2) mean root diamete（mm） Total Root Volume（cm3）

Low 17.1920±3.7470c 1.9814±0.5618c 0.3607±0.0789b 0.0212±0.0039b

Mid 20.5585±3.4703b 3.3160±0.5553b 0.4111±0.0622ab 0.0455±0.0076a

High 26.8330±5.1689a 3.9784±0.2801a 0.4775±0.0640a 0.0470±0.0023a

In response, we expanded on the potential reasons for the negative correlation between root density and shear strength, as shown in Section 4.2, lines 594-599.

Q7：Reference [16] (Wischmeier, 1978) is a classic model, but recent advances in the field (e.g., studies on soil erosion in reservoir fluctuation zones post-2020) are not cited. Some references (e.g., [65]) are cited redundantly.

R7：Thank you for your valuable feedback. In response to your suggestions, we have revised the introduction to include recent advancements in the field. Specifically, we have incorporated relevant studies by Kinnell (2015), Thomas and Joseph et al. (2018), and Duan and Bai et al. (2020) into the second paragraph of the introduction to enhance the timeliness and relevance of our literature review (Kinnell, 2015; Thomas and Joseph et al., 2018; Duan and Bai et al., 2020). Additionally, we have streamlined the reference list by removing redundant citations. However, a small number of seminal but older references have been retained due to their foundational role in the topic and the current lack of modern alternatives.

Grammatical Issues

Q8: Formula symbols like "DSA 0.25" are inconsistently mixed with Chinese text in the Abstract. Some paragraphs (e.g., paragraph 3 of the Abstract) contain grammatical errors, such as the misspelling "t is sisnificntly positively correlated".

R8: Thank you for your valuable feedback. We have revised the manuscript accordingly. Specifically, the notation "DSA 0.25" has been standardized to "DSA₀.₂₅" using proper subscript formatting. Additionally, a spelling error has been corrected in the sentence previously stating "t is sisnificntly positively correlated", which now reads "it is significantly positively correlated."

Q9: Terms like "colluvial slope soils" (Abstract) and "colluvial zone soil" (Abstract) are used inconsistently. "Erosion coefficient" and "erodibility factor K" are used interchangeably, causing confusion.

R9: Thank you for your valuable feedback. In accordance with your suggestions, we have carefully revised the manuscript to ensure terminological consistency. Specifically, the terms "colluvial zone soil" and "erodibility factor K" have been standardized and used consistently throughout the article.

Q10:The last paragraph of the Introduction ("Over all the response of soil detachment...") has overly complex sentence structure and grammatical gaps (e.g., missing subject before "are in need of further study")

R10: Thank you for your valuable feedback. We have carefully revised the manuscript accordingly. Specifically, the original sentence “Over all the response of soil detachment...” has been restructured into two clearer sentences: “The response of soil detachment to different combinations of soil and root system properties has not been fully quantified. More research is needed in this area.” Additionally, the sentence “it was speculated that...” has been corrected to “It is speculated that...” to maintain appropriate academic tone and grammatical consistency.

Errors in Methods, Knowledge, and Techniques

Q11: The sentence "Over all the response of soil detachment..." was changed into two sentences: "The response of soil detachment to different combinations of soil and root system properties has not been fully quantified. More research is needed in this area." Additionally, the phrase "it was speculated that root traits may mediate the impact of plant diversity on riparian soil stability." was revised to "It is speculated that root traits may mediate the impact of plant diversity on riparian soil stability."

R11: Thank you very much for your thoughtful comments regarding the methodological considerations in our study.

In response to your feedback, we would like to clarify the rationale behind the selection of the K factor formula. As noted by Zhang et al. (2008), the direct application of the three original K factor estimation methods under Chinese conditions resulted in notable overestimations. Accordingly, Zhang and colleagues recalibrated the formula to improve its suitability for regional soil conditions, and it is this revised formula that has been adopted in our study.

Regarding the logarithmic base used in the formula, we acknowledge that Shirazi et al. (1988) 【Shirazi, M. A. and L. Boersma, et al. (1988). "A unifying quantitative analysis of soil texture: improvement of precision and extension of scale." Soil Science Society of America Journal 52 (1): 181-190.】employed base 10 in their original work. However, subsequent applications—including recent studies such as Wu et al. (2023)—have consistently utilized the natural logarithm 【log(GMD)】 in related contexts. 【Wu, T. and Y. Zhang, et al. (2023). "Factors affecting the stability of soil aggregates of plinthosols in the middle reaches of the Yangtze River." Catena 228: 107159.】In order to maintain consistency with prevailing literature and ensure broader comparability of results, we retained the natural logarithm in our formula. A clarifying note has been included beneath the equation to explicitly indicate the use of base 10 where relevant, thereby preserving both mathematical integrity and explanatory clarity.

Q12: In Section 3.1, the statement "root zone soil bulk density decreased by 24.39% compared to non-root zone" lacks indication of statistical significance (e.g., p-value from t-test or ANOVA). Merely describing numerical differences does not meet academic standards.

R12: Thank you for your valuable feedback. In response to your suggestions, we have added descriptions of significant differences throughout the Results section. For instance, the following statement has been included: “At each elevation, the bulk density of the root soil was significantly lower than that of the rootless soil (P < 0.05) at both low and high elevations, with reductions of 24.39% and 24.86%, respectively.”

Additionally, we have included asterisks (*) in Figure 5.

Q13: In Section 2.3, the definition of "PAD (X12)" as "aggregate of destruction" is unclear regarding its relationship to "aggregate destruction rate". Additionally, units for "DSA 0.25" and "WSA₀.₂₅" in formula (4) are missing (should be mass percentage).

R13: Thank you for your valuable feedback.In response, we have clarified the terminology and units throughout the manuscript. Specifically, the term PAD is now consistently defined as "aggregate destruction rate" in all relevant table captions and equation descriptions. Additionally, units have been explicitly added for DSA₀.₂₅ and WSA₀.₂₅ as (wt%) wherever applicable.

Comprehensive Revision Suggestions

Thank you very much for your thorough review and valuable suggestions. We have carefully revised the manuscript accordingly to address your comments. The main improvements are summarized as follows:

In the Discussion section, the content has been reorganized along the logical chain of “water level fluctuations → changes in soil properties → anti-scour response” to improve clarity and avoid redundancy. Units have been consistently added for key parameters: K as (t·h·MJ⁻¹·mm⁻¹), DSA₀.₂₅ as (wt%), and WSA₀.₂₅ as (wt%).

In Section 2.4, the KMO test and Bartlett’s test of sphericity have been included to strengthen the validation of the Principal Component Analysis (PCA). Additionally, root-related data have been supplemented where appropriate.

In the second paragraph of the Introduction, we have incorporated the most recent research advances in the field and removed the majority of redundant references. However, a limited number of classical references were retained due to the unavailability of contemporary alternatives for foundational models.

Terminology has been standardized throughout the manuscript—e.g., “DSA₀.₂₅”, “soil erodibility factor K”, and “colluvial zone soil”—and grammatical errors have been corrected. Furthermore, significance testing statements have been added where applicable, and PAD is now explicitly defined as "aggregate destruction rate" in all relevant sections.

We believe these revisions have significantly improved the quality and clarity of the manuscript and thank you once again for your constructive input.

Special thanks to you for your good comments.

We have made every effort to improve the manuscript and have implemented some changes within it. These alterations are designed to enhance the clarity and presentation without affecting the core content and structure of the paper. Instead of listing the changes here, we have highlighted them in the revised manuscript for your direct reference.

We extend our sincere gratitude to the Editors and Reviewers for their diligent work and constructive feedback

---

## [Editor Report · Decision Letter 1]

28 Sep 2025

Dear Dr. Xiao,

Thank you for submitting your manuscript to PLOS ONE. After careful consideration, we feel that it has merit but does not fully meet PLOS ONE’s publication criteria as it currently stands. Therefore, we invite you to submit a revised version of the manuscript that addresses the points raised during the review process.

We look forward to receiving your revised manuscript.

Kind regards,

Somayeh Soltani-Gerdefaramarzi, Ph. D.

Academic Editor

PLOS ONE
---

## [Author Response · Author response to Decision Letter 2]

10 Oct 2025

Dear PLOS ONE Editorial Office,

Thank you for your email and for providing us with the opportunity to revise our manuscript.

We have carefully addressed the points raised. The key revisions are as follows:

1. We have updated the reference list, removing a number of non-essential citations to enhance the focus and relevance of the literature presented.

2. As requested, we have reformatted and resized all figures using the PACE tool to ensure they now meet the journal's specifications.

We believe these revisions have significantly improved the manuscript and hope it now meets the standards for publication in PLOS ONE.

Sincerely,

Yours sincerely,

Henglin Xiao & Pengcheng Wang

10 October, 2025

Corresponding author: Prof. Henglin Xiao

E-mail: xiaohenglin@hbut.edu.cn

---

## [Decision Letter · Decision Letter 2]

29 Oct 2025

Quantitative evaluation of soil anti-erodibility in the fluctuation zones of rooted soil in a large reservoir, southwest of China

PONE-D-25-21716R2

Dear Dr. Xiao,

We’re pleased to inform you that your manuscript has been judged scientifically suitable for publication and will be formally accepted for publication once it meets all outstanding technical requirements.

Kind regards,

Somayeh Soltani-Gerdefaramarzi, Ph. D.

Academic Editor

PLOS ONE

Additional Editor Comments (optional):

Reviewers' comments:

Reviewer's Responses to Questions

**Comments to the Author**

Reviewer #1: All comments have been addressed

2. Is the manuscript technically sound, and do the data support the conclusions?

Reviewer #1: Yes

3. Has the statistical analysis been performed appropriately and rigorously?

Reviewer #1: Yes

4. Have the authors made all data underlying the findings in their manuscript fully available?

Reviewer #1: Yes

5. Is the manuscript presented in an intelligible fashion and written in standard English?

Reviewer #1: Yes

Reviewer #1: I have carefully evaluated the revised version of the manuscript entitled “Quantitative evaluation of soil anti-erodibility in the fluctuation zones of rooted soil in a large reservoir, southwest of China.” After reviewing the authors’ responses and the changes made during the two rounds of revision, I am satisfied that all previous concerns raised by the reviewers have been thoroughly addressed.

The authors have substantially improved the clarity, scientific rigor, and presentation of the manuscript. In particular:

1) The literature review has been refined to better contextualize the study within current research.

2) Figures have been reformatted and improved in quality, ensuring clear visualization and interpretation of the results.

3) The methods and results are now presented with greater clarity and logical progression, enhancing the overall readability.

4) The conclusions are well supported by the data and are appropriately framed within the limitations of the study.

The study provides valuable empirical evidence on soil erosion resistance mechanisms in reservoir fluctuation zones, focusing on the influence of plant roots and soil physicochemical properties. This is an important and timely contribution to the field of soil conservation and ecological restoration in riparian systems. The comprehensive analytical approach, including the use of principal component analysis and the development of a soil erosion resistance evaluation model, strengthens the scientific merit of the work and increases its relevance to both researchers and practitioners.

**Do you want your identity to be public for this peer review?** For information about this choice, including consent withdrawal, please see our Privacy Policy

Reviewer #1: **Yes**

---

## [Editor Report · Acceptance letter]

PONE-D-25-21716R2

PLOS One

Dear Dr. Xiao,

I'm pleased to inform you that your manuscript has been deemed suitable for publication in PLOS One. Congratulations! Your manuscript is now being handed over to our production team.

Kind regards,

on behalf of

Dr. Somayeh Soltani-Gerdefaramarzi

Academic Editor

PLOS One